# Filamin, a synaptic organizer in *Drosophila*, determines glutamate receptor composition and membrane growth

**GaYoung Lee[1,2], Thomas L Schwarz[1,2]***

[1]The F.M. Kirby Neurobiology Center, Boston Children's Hospital, Boston, United States; [2]Department of Neurobiology, Harvard Medical School, Boston, United States

**Abstract** Filamin is a scaffolding protein that functions in many cells as an actin-crosslinker. FLN90, an isoform of the Drosophila ortholog Filamin/cheerio that lacks the actin-binding domain, is here shown to govern the growth of postsynaptic membrane folds and the composition of glutamate receptor clusters at the larval neuromuscular junction. Genetic and biochemical analyses revealed that FLN90 is present surrounding synaptic boutons. FLN90 is required in the muscle for localization of the kinase dPak and, downstream of dPak, for localization of the GTPase Ral and the exocyst complex to this region. Consequently, Filamin is needed for growth of the subsynaptic reticulum. In addition, in the absence of filamin, type-A glutamate receptor subunits are lacking at the postsynapse, while type-B subunits cluster correctly. Receptor composition is dependent on dPak, but independent of the Ral pathway. Thus two major aspects of synapse formation, morphological plasticity and subtype-specific receptor clustering, require postsynaptic Filamin.

**\*For correspondence:** Thomas. Schwarz@childrens.harvard.edu

**Competing interests:** The authors declare that no competing interests exist.

## Introduction

Proper postsynaptic function depends on appropriate localization of receptors and signaling molecules. Scaffolds such as PSD-95/SAP90 and members of the Shank family are critical to achieving this organization. While usually without intrinsic enzymatic activity, scaffolds recruit, assemble, and stabilize receptors and protein networks through multiple protein-protein interactions: they can bind to receptors, postsynaptic signaling complexes, and the cytoskeleton at the postsynaptic density (*Sheng and Kim, 2011*). Mutations in these proteins are associated with neuropsychiatric disorders. While we are beginning to understand synapse assembly, much remains to be investigated.

The *Drosophila* larval neuromuscular junction (NMJ) is a well-studied and genetically accessible glutamatergic synapse. Transmission is mediated by AMPA-type receptors, and several postsynaptic proteins important for its development and function have related proteins at mammalian synapses, including the PDZ-containing protein Discs-Large (DLG) and the kinase Pak (*Ataman et al., 2006*; *Collins and DiAntonio, 2007*; *Hayashi et al., 2004*; *Kreis and Barnier, 2009*; *Penzes et al., 2003*). In addition, the postsynaptic membrane forms deep invaginations and folds called the subsynaptic reticulum (SSR), which are hypothesized to create subsynaptic compartments comparable to dendritic spines. Recently, we found that the SSR is a plastic structure whose growth is regulated by synaptic activity (*Teodoro et al., 2013*). This phenomenon may be akin to the use-dependent morphological changes, such as growth of dendritic spines, that occur postsynaptically in mammalian brain. The addition of membrane and growth of the SSR requires the exocyst complex to be recruited to the synapse by the small GTPase Ral; the SSR fails to form in *ral* mutant larvae.

Moreover, the localization of Ral to a region surrounding synaptic boutons is likely to direct the selective addition of membrane to this domain. Ral thus provided a tractable entry point for better understanding postsynaptic assembly. The mechanism for localizing the Ral pathway, however, was unknown. In the present study, we determined that Ral localization is dependent on *cheerio*, a gene encoding filamin, which we now show to be critical for proper development of the postsynapse.

Filamin is a family of highly conserved protein scaffolds with a long rod-like structure of Ig-like repeats. With over 90 identified binding partners, some of which are present also at the synapse, mammalian filamin A (FLNA) is the most abundant and commonly studied filamin (*Feng and Walsh, 2004*). Filamin can bind actin and has received the most attention in the context of actin cytoskeletal organization (*Nakamura et al., 2011*). *Drosophila* filamin, encoded by the gene *cheerio* (*cher*), shares its domain organization and 46% identity in amino acid sequence with human FLNA. *Drosophila* filamin has a well-studied role in ring canal formation during oocyte development, where it recruits and organizes actin filaments (*Li et al., 1999*; *Robinson et al., 1997*; *Sokol and Cooley, 1999*). We now show that filamin has an essential postsynaptic role at the fly NMJ. We find that an isoform of this scaffold protein that lacks the actin-binding domain acts via dPak to localize GluRIIA receptors and Ral; filamin thereby orchestrates both receptor composition and membrane growth at the synapse.

## Results

### Ral localization and postsynaptic membrane addition require filamin

Immunolocalization of Ral expressed in muscles of wild type animals reveals a distinct halo around each synaptic bouton, a distribution resembling that of the subsynaptic reticulum (*Teodoro et al., 2013*; *Figure 1*). This distribution (hereafter subsynaptic Ral) uniformly surrounds the bouton and is therefore distinct from the more punctate distribution of glutamate receptors, which are restricted to the membranes immediately opposite active zones (*Petersen et al., 1997*). Wild type Ral and Ral mutants locked in either the GTP or GDP-bound states share this subsynaptic distribution (*Figure 1B–D*, control). To determine factors responsible for Ral localization, we expressed in larval muscle RNAi directed against candidate proteins, including filamin (*Ohta et al., 1999*), that are reported in the literature or in proteomic databases to interact with Ral. RNAi against filamin prevented the concentration around boutons of expressed Ral transgenes (*Figure 1B, C, and F*). To verify the RNAi phenotype, we used a combination of existing alleles: $cher^{Q1415sd}$, a truncation that behaves genetically as a null allele, and $cher^{\Delta 12.1}$, a deficiency lacking the entire coding region (*Li et al., 1999*; *Sokol and Cooley, 1999*). $cher^{Q1415sd}/cher^{\Delta 12.1}$ larval muscles, like those expressing filamin RNAi, lacked subsynaptic Ral (*Figure 1D and F*). Ral was still present in the muscle cytoplasm, and there was no change in total protein levels of Ral. Muscles lacking filamin still developed grossly normal and the innervation of muscles 6/7 had an architecture and bouton number comparable to controls (*Figure 1—figure supplement 1A–D*).

During SSR growth, activated Ral recruits the exocyst, a membrane-tethering complex that can be visualized by immunostaining for Sec5, a central component of the complex (*Teodoro et al., 2013*). Overexpressing GTP-locked Ral ($Ral^{GTP}$) in muscles activates subsynaptic exocyst recruitment. Concurrent knockdown of filamin in the muscles, however, prevented exocyst recruitment by $Ral^{GTP}$ (*Figure 1E and F*). Together, these data support the conclusion that filamin is required for localizing the pathway for activity-dependent postsynaptic membrane growth.

### Filamin regulates postsynaptic membrane architecture

Because the absence of filamin prevented the synaptic targeting of Ral and subsequent recruitment of the exocyst, we suspected that SSR formation would also be compromised. To test this, we first used an antibody against Syndapin, an SSR marker (*Kumar et al., 2009a*). Syndapin immunoreactivity was greatly reduced with both muscle expression of filamin RNAi and the $cher^{Q1415sd}/cher^{\Delta 12.1}$ genotype (*Figure 2A and B*). We confirmed by electron microscopy that the change in Syndapin reflected a reduction in SSR size (*Figure 2C and D*). Many $cher^{Q1415sd}/cher^{\Delta 12.1}$ boutons completely lacked the SSR: the presynaptic bouton only faced a single layer of membrane with appropriate postsynaptic densities opposite the active zones. Others had SSR but it extended less deeply into the muscle than in controls and lacked the characteristic complexity of membrane folds. Moreover,

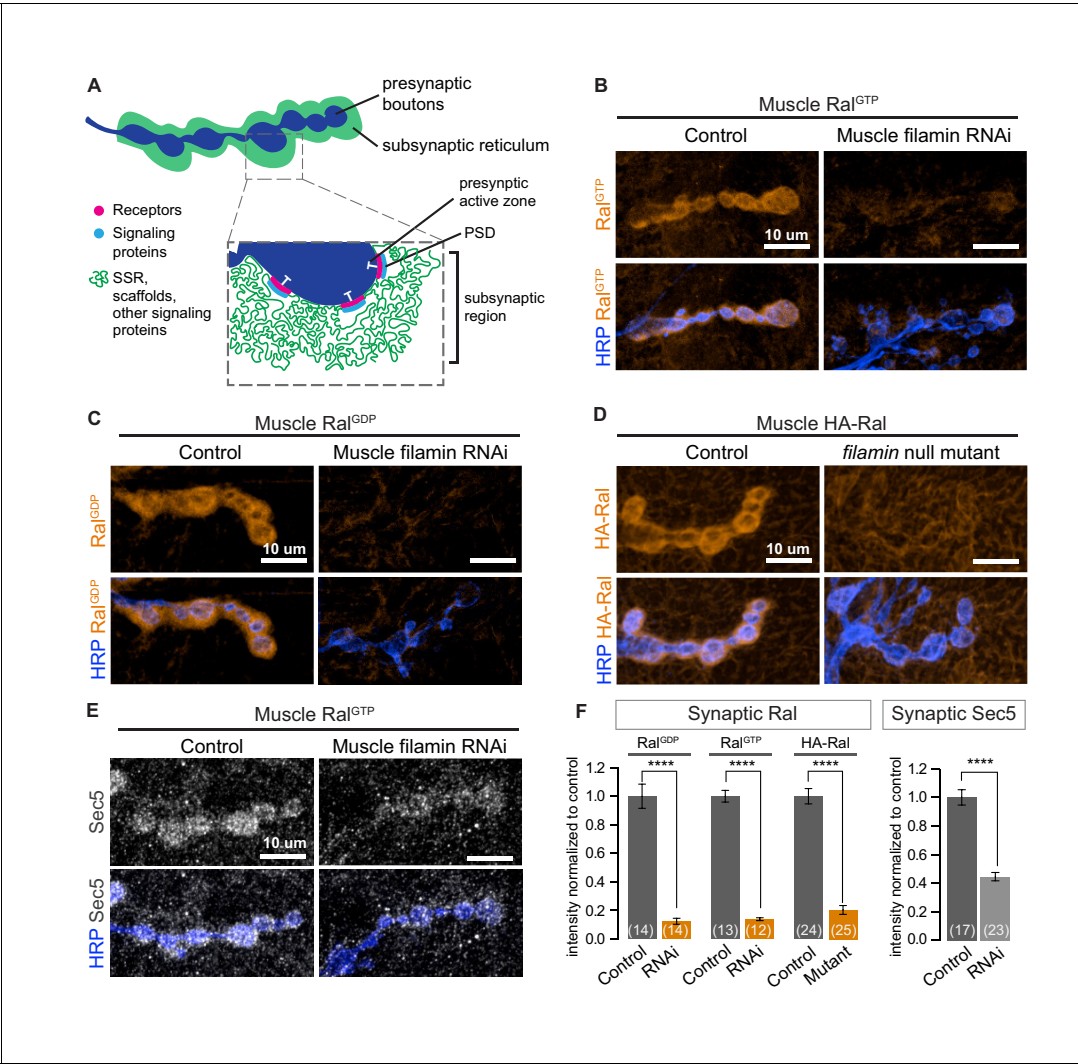

**Figure 1.** Subsynaptic localization of Ral and Sec5 requires muscle filamin. (**A**) Schematic cross-section of a larval neuromuscular junction (NMJ) illustrating the locations of postsynaptic components. The presynaptic active zone, marked by an electron-dense T-bar (white), faces the postsynaptic density (PSD) that contains glutamate receptors (magenta) and associated signaling molecules (blue). The extensively folded postsynaptic membrane, the subsynaptic reticulum (SSR, green), occupies a subsynaptic region that extends into the muscle and surrounds the boutons. The SSR contains membrane curvature proteins, including Syndapin, and signaling molecules, including Ral. Proteins in the SSR are not uniformly distributed throughout; some components, including Dlg, are restricted to the more superficial layers (*Koles et al., 2015*). (**B–C**) Confocal images of NMJs immunostained for constitutively-active Ral$^{GTP}$ (**B**) or GDP-locked Ral (**C**) expressed in muscles of control larvae (*G14-Gal4/+;UAS-Ral$^{GTP}$/+* and *G14-Gal4/UAS-Ral$^{GDP}$;+/+*) and larvae with muscle co-expression of RNAi against filamin (*G14-Gal4/+;UAS-Ral$^{GTP}$/UAS-filamin RNAi* and *G14-Gal4/UAS-Ral$^{GDP}$;UAS-filamin RNAi/+*). To detect Ral, an anti-Ral antibody that does not noticeably detect endogenous Ral (*Teodoro et al., 2013*) is used. In this and all subsequent figures, presynaptic nerve terminals are labeled using an antibody against neuronal membrane marker HRP (horseradish peroxidase). (**D**) Confocal images of NMJs immunostained for HA-Ral expressed in the postsynaptic muscles of control (*UAS-HA-Ral/G14-Gal4;cher$^{Δ12.1}$/+*) and filamin null larvae (*UAS-HA-Ral/G14-Gal4; cher$^{Δ12.1}$/cher$^{Q1415sd}$*). (**E**) Confocal images of NMJs immunostained for the exocyst component Sec5 which is recruited to the subsynaptic region by muscle expression of Ral$^{GTP}$ in control larvae (*UAS-Ral$^{GTP}$,MHCGS/+*) but not upon coexpression of RNAi against filamin (*UAS-Ral$^{GTP}$,MHCGS/UAS-filamin RNAi*). (**F**) Quantification of mean synaptic fluorescence intensities for the genotypes in (**B–E**), each normalized to their control. The images selected here and in all subsequent figures reflect the mean values for that genotype and, unless otherwise specified, are shown as z-stacks with maximum-intensity projection. For all quantifications, values are background-subtracted and normalized to control. Scale bar: 10 μm. Number of NMJs quantified is indicated in each graph; number of animals quantified in this and subsequent figures are indicated in a table in the Materials and methods section. Statistical significance was determined using two-tailed unpaired t test, ****p<0.0001. All error bars indicate ± SEM.

The following figure supplement is available for figure 1:

**Figure supplement 1.** Loss of filamin impairs synaptic localization of Ral independent of Ral protein levels or muscle integrity.

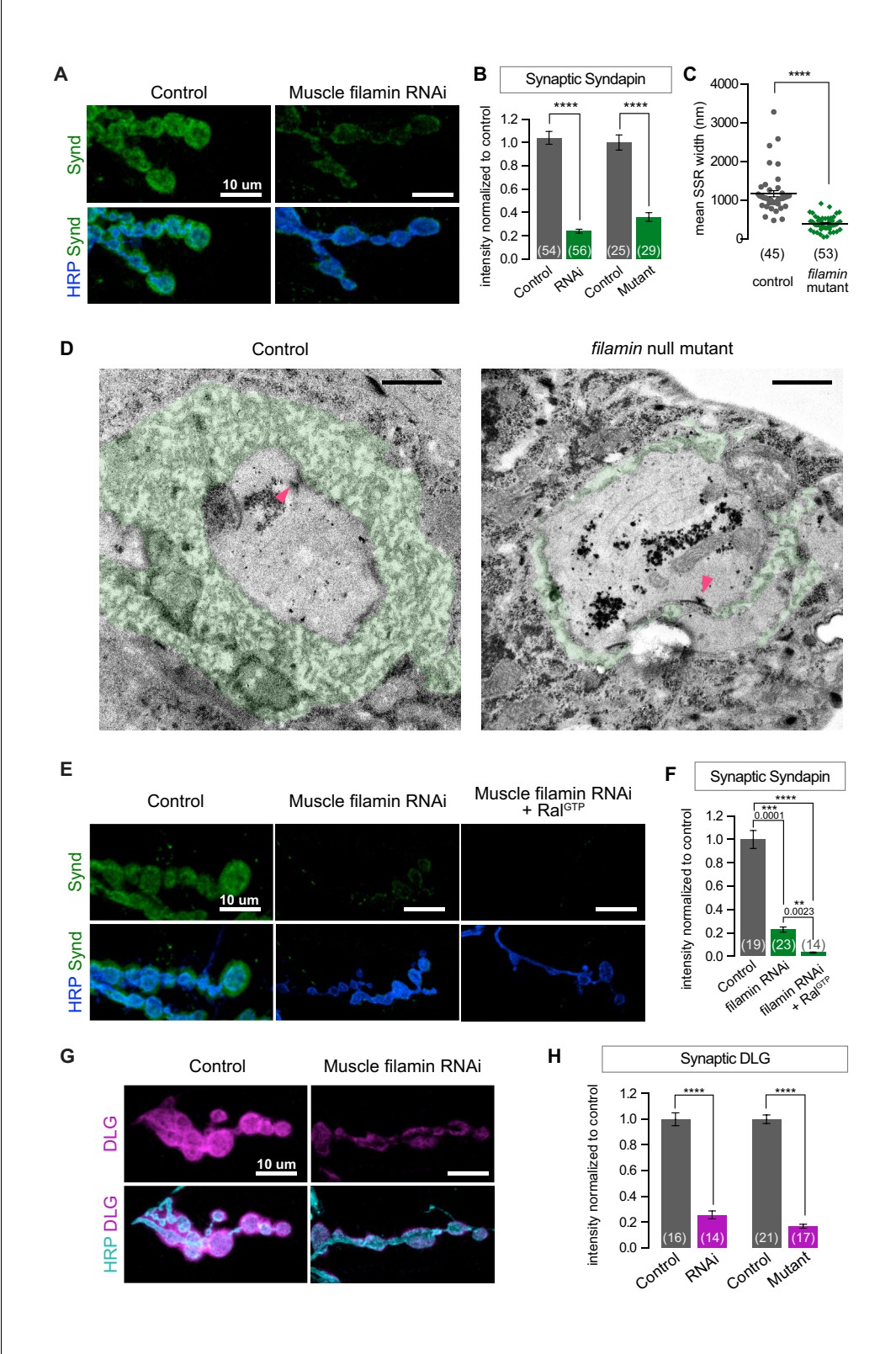

**Figure 2.** SSR formation depends on filamin. (**A**) Confocal images of NMJs immunostained for Syndapin, an SSR marker, at NMJs of a control (*G14-Gal4/+*) larva and one with muscle-specific knockdown of filamin (*G14-Gal4/+;UAS-filamin RNAi/+*). (**B**) Quantification of Syndapin immunofluorescence intensities in the following genotypes: *G14-Gal4/+* (RNAi control), *G14-Gal4/+;UAS-filamin RNAi/+* (RNAi), *cher^{Δ12.1}/+* (control for mutant), *cher^{Δ12.1}/cher^{Q1415sd}* (mutant). (**C,D**) Quantification of SSR width and representative electron micrographs in control (*cher^{Δ12.1}/+*) and *filamin* mutant (*cher^{Δ12.1}/*
*Figure 2 continued on next page*

Figure 2 continued

$cher^{Q1415sd}$). Micrographs show cross-sections of individual boutons and their surrounding SSR, which has been highlighted in green. Note the complex intertwining in control muscles of electron-dense muscle cytoplasm and translucent extracellular space that characterizes the SSR. Arrowheads indicate active zones, with characteristic T-bars. (E,F) Confocal images and quantification of Syndapin immunoreactivity at the NMJ for controls (G14-Gal4/+;+/+) and larvae with muscle-specific filamin knockdown with and without co-expression of Ral$^{GTP}$ (G14-Gal4/+;UAS-filamin RNAi and G14-Gal4/+;UAS-filamin RNAi/UAS-Ral$^{GTP}$). (G) Confocal images of Discs-Large (DLG) immunofluorescence at NMJs of controls (G14-Gal4/+;+/+) and larvae with muscle-specific filamin knockdown (G14-Gal4/+;UAS-filamin RNAi). (H) Quantification of mean synaptic DLG immunofluorescence for the genotypes as in (B). Scale bar in (A), (E), and (G): 10 μm; in (D): 1 μm. Number of NMJs quantified is indicated in each graph except (C), where n= number of individual boutons. For all quantifications except in (F), two-tailed unpaired t test was performed for statistical significance. Multiple comparisons in (F) were performed using Kruskal-Wallis test with Dunn's multiple comparisons test. ****p<0.0001; other p values are indicated on each graph. Error bars indicate ± SEM.

The following figure supplement is available for figure 2:

**Figure supplement 1.** Synaptic DLG localization requires muscle expression of Ral.

even with muscle overexpression of Ral$^{GTP}$, a genotype characterized by overgrowth of the SSR in a wild type background (*Teodoro et al., 2013*), no restoration of the SSR was detected; Syndapin immunoreactivity was not increased above the levels in muscles expressing filamin RNAi alone (*Figure 2E and F*). We conclude that filamin enables SSR formation via recruitment of subsynaptic Ral.

Discs-Large (DLG) is another postsynaptic constituent required for SSR formation. A fly homologue of PSD-95/SAP97/PSD-93, it is a member of the membrane-associated guanylate kinase (MAGUK) family of scaffolding proteins and is a key player in NMJ development (*Budnik et al., 1996*; *Lahey et al., 1994*). It is present both within presynaptic boutons and in the portion of the SSR closest to the bouton (*Koles et al., 2015*). Knocking down filamin disrupted the localization of DLG to the postsynapse, as did the $cher^{Q1415sd}$/$cher^{Δ12.1}$ allelic combination (*Figure 2G and H*). This phenotype is likely a consequence of the mislocalization of Ral, as NMJs lacking Ral also have reduced levels of DLG (*Figure 2—figure supplement 1A–C*).

## The short FLN90 isoform is required for SSR formation, while the long isoform is dispensable

In *Drosophila* oocytes, the $cher^1$ allele disrupts actin architecture and fails to recruit ring canal proteins (*Li et al., 1999*; *Robinson et al., 1997*). We therefore expected that $cher^1$ phenotypes would match those of filamin RNAi and the $cher^{Q1415sd}$ allele. Contrary to our expectations, in $cher^1$/$cher^{Δ12.1}$ larvae Syndapin was present surrounding boutons at nearly wild type levels (*Figure 3A*). Subsynaptic Ral immunoreactivity was significantly decreased, but to a far lesser extent than with the $cher^{Q1415sd}$ allele (*Figure 3B*). At muscle 4, a muscle where bouton size and morphology are less variable than at muscles 6 and 7 (shown in *Figure 3A and B*), the $cher^1$ phenotype was even weaker (*Figure 3C*).

What could account for the discrepancy between $cher^1$ and $cher^{Q1415sd}$? The *cheerio* locus produces two transcripts that give rise to two filamin isoforms: (1) a 7.5 kb transcript produces a full-length 240 kDa protein (FLN240, the 'long isoform') and (2) a 3 kb transcript produces a 90 kDa protein (FLN90, the 'short isoform'). The latter is derived from the 3' end of the 7.5 kb transcript, such that FLN90 comprises the C-terminal portion of FLN240 (*Figure 3D*) (*Li et al., 1999*; *Sokol and Cooley, 1999*). The RNAi initially used to identify filamin in our study targets this shared region, thereby knocking down both isoforms. In wild-type whole larval lysates, an antibody directed against the C terminus shared by both isoforms (*Li et al., 1999*) detected bands corresponding to both isoforms. The specificity of the immunoreactive bands was confirmed by their loss in $cher^{Q1415sd}$/$cher^{Δ12.1}$ larvae (*Figure 3E*). In $cher^1$/$cher^{Δ12.1}$ (abbreviated as $cher^1$), however, the FLN90 band was still present although FLN240 was not detectable with either a C-terminus- (*Figure 3E*) or an N-terminus-directed antibody (*Külshammer and Uhlirova, 2013*). The continued presence of FLN90 is consistent with the molecular characterization of $cher^1$ in which a P-element disrupts only the 7 kb transcript (*Sokol and Cooley, 1999*). A point mutation in $cher^{Q1415sd}$ disrupts splicing of both transcripts and consequently in $cher^{Q1415sd}$/$cher^{Δ12.1}$ (abbreviated as $cher^{Q1415sd}$), FLN90 was barely detectable

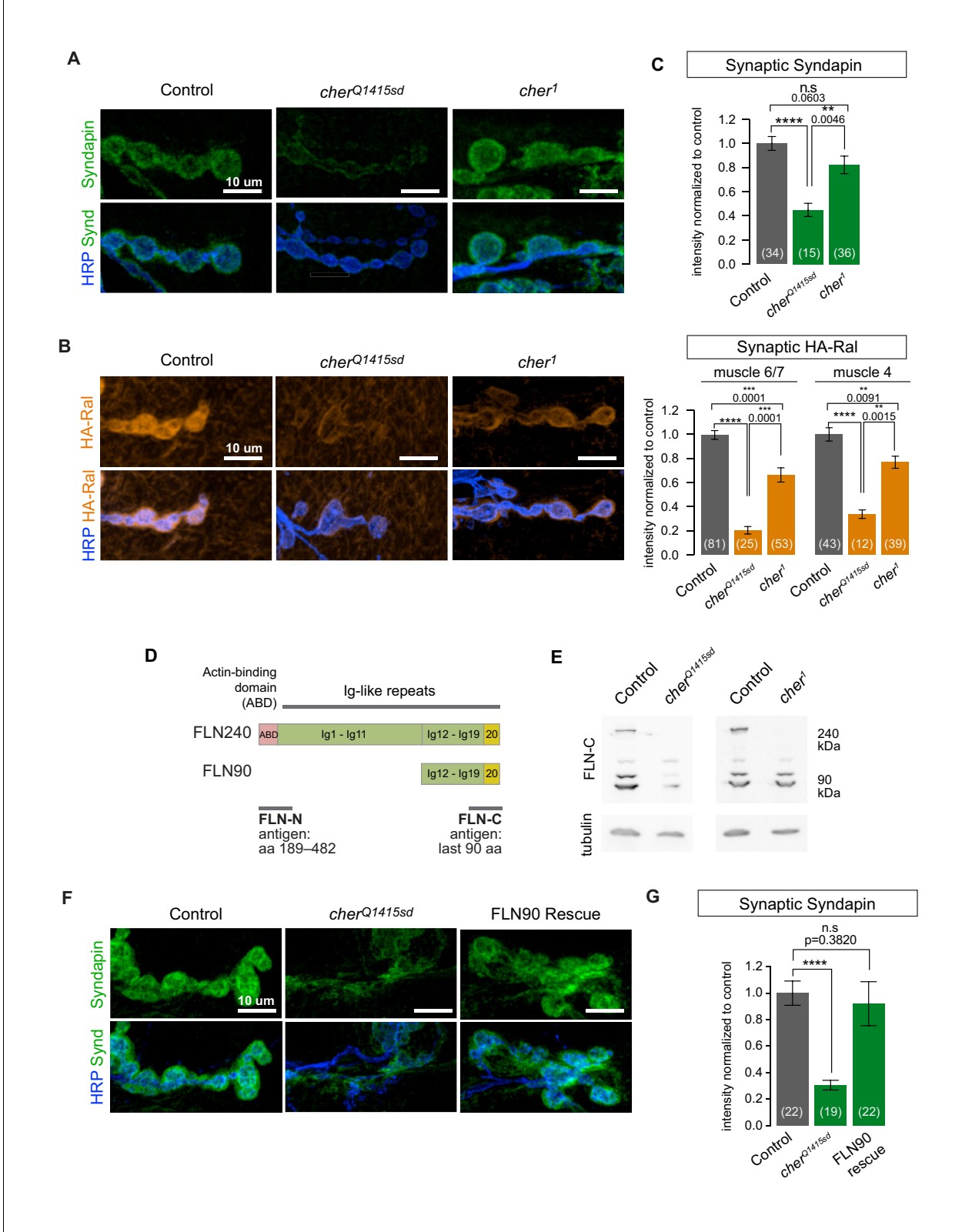

**Figure 3.** The short FLN90 isoform recruits Ral and promotes SSR formation. (**A**) Confocal images of Syndapin immunoreactivity in the genotypes $cher^{\Delta 12.1}/+$ (control), $cher^{\Delta 12.1}/cher^{Q1415sd}$ (cher$^{Q1415sd}$), and $cher^{\Delta 12.1}/cher^1$ (cher$^1$). (**B**) Anti-HA immunoreactivity in the genotypes *UAS-HA-Ral/G14-Gal4; $cher^{\Delta 12.1}/+$* (control), *UAS-HA-Ral/G14-Gal4; $cher^{\Delta 12.1}/cher^{Q1415sd}$* (cher$^{Q1415sd}$), and *UAS-HA-Ral/G14-Gal4; $cher^{\Delta 12.1}/cher^1$* (cher$^1$). (**C**) Quantification of mean synaptic Syndapin and HA-Ral immunofluorescence for the genotypes in (**A,B**). Quantification of HA-Ral is given for NMJs on

*Figure 3 continued on next page*

*Figure 3 continued*

muscle 4 and muscles 6/7. (**D**) Domain organization of the two filamin isoforms. The FLN90 isoform consists of the C-terminal region of the FLN240 isoform. The location of the actin-binding domain (ABD) is shown and positions of Ig domains are numbered. Also indicated are the antigens for the two filamin antibodies used in this study. (**E**) Western blots of third-instar larval lysates probed with the FLN-C antibody. FLN240 and FLN90 (which appears as a doublet) are both present in control heterozygous larvae. In cher$^{Q1415sd}$ (cher$^{Δ12.1}$/cher$^{Q1415sd}$) both isoforms are almost undetectable; in cher$^1$ (cher$^{Δ12.1}$/cher$^1$) the short isoform remains. Tubulin immunoreactivity is shown as a loading control. (**F**) Syndapin immunoreactivity in the genotypes *G14-Gal4/+;cher$^{Q1415sd}$/+* (control), *G14-Gal4/+;cher$^{Q1415sd}$/cher$^{Δ12.1}$* (cher$^{Q1415sd}$), and *G14-Gal4/UAS-HA-FLN90;cher$^{Q1415sd}$/cher$^{Δ12.1}$* (rescue). (**G**) Quantification of mean synaptic Syndapin immunofluorescence for the genotypes (**F**). Scale bars: 10 um. Number of NMJs quantified is indicated in each graph. Multiple comparisons in (**C**) and (**G**) performed using Kruskal-Wallis test with Dunn's multiple comparisons test. ****p<0.0001; other p values are indicated. Error bars indicate ± SEM.

The following figure supplement is available for figure 3:

**Figure supplement 1.** Evidence confirming selective rescue of the short FLN90 isoform.

(***Figure 3E***) (the remaining protein is probably due to residual maternal protein) and only a truncated N-terminal fragment of FLN240 is made (***Li et al., 1999***; ***Sokol and Cooley, 1999***, ***2003***). This difference between the alleles correlates with their different phenotypes at the NMJ and suggests that the FLN90 isoform that persists in cher$^1$ is sufficient to permit SSR formation. We addressed this possibility also by genetic rescue. When only the FLN90 isoform was expressed in muscles of filamin-null larvae lacking both isoforms, levels of Syndapin immunoreactivity were restored (***Figure 3F and G***, and ***Figure 3—figure supplement 1***). Therefore, FLN90 expression in the muscle is sufficient to drive SSR growth.

To determine whether the short isoform is localized to the synapse, we visualized the pattern of endogenous filamin expression with two anti-filamin antibodies. One antibody (***Külshammer and Uhlirova, 2013***; hereby referred to as anti-FLN-N) is directed against residues 189–482, which form the actin-binding domain and the first two Ig-like repeats on the N terminus; therefore, it detects only the long isoform. The other antibody (***Li et al., 1999***; hereafter anti-FLN-C) is directed against 90 residues at the C terminus and detects both isoforms (***Figure 3D***). FLN-N immunoreactivity was apparent in nerve, trachea, and glia, and in puncta in presynaptic boutons (***Figure 4A***). FLN-C immunoreactivity was diffuse throughout muscle cytoplasm, in trachea, and within synaptic boutons, but markedly greater surrounding the boutons in a manner similar to Ral and Syndapin. The specificity of this subsynaptic signal was confirmed by knocking down filamin in the muscles, which resulted in loss of the postsynaptic but not the presynaptic signal (***Figure 4B***).

The ability to detect the postsynaptic signal with FLN-C but not FLN-N indicated that FLN90 is the isoform present postsynaptically. This was also confirmed by biochemical analysis. Lysates of larval pelts, which are composed primarily of body wall muscles, contained almost exclusively the short isoform; in contrast, the long isoform was predominant in lysates of larval brain. In addition, RNAi expression in muscle removed the FLN90 band but not the long isoform which is likely contributed by other cell types in the pelts, while RNAi expression in the nervous system had the reverse effect (***Figure 4C and D***). Similarly, at cher$^{Q1415sd}$/cher$^{Δ12.1}$ NMJs, the filamin signal was strongly reduced, but in cher$^1$/cher$^{Δ12.1}$ NMJs, filamin immunoreactivity still surrounded the boutons (***Figure 4E***). Moreover, an epitope-tagged short isoform (GFP-FLN90), when expressed in muscles, surrounds the synaptic boutons (***Figure 4F***). Together, multiple lines of evidence indicate that the postsynaptic signal is mostly, and perhaps entirely, derived from the short FLN90 isoform and that this isoform is sufficient to achieve SSR formation.

Filamin is also present in muscles earlier during NMJ development (***Figure 4—figure supplement 1***). In late-stage embryos and first-instar larvae it is diffuse in the muscles, including at sites of nerve contact; this continues into the second-instar stage although FLN90 is noticeably more concentrated around the boutons at this time. Thus the accumulation of filamin in the vicinity of boutons occurs in parallel with the expansion of the SSR. To determine whether filamin accumulation required the growth of the SSR, we examined FLN90 expression in genetic *ral* null (*ral$^{G0501}$*) NMJs, which lack the SSR due to the loss of Ral-mediated membrane growth (***Teodoro et al., 2013***). Synaptic FLN90 localization was unaffected by the absence of Ral; it continued to surround the boutons of third instar larvae, as in wild type (***Figures 4F*** and ***5***). Thus, the finding that SSR growth depends on

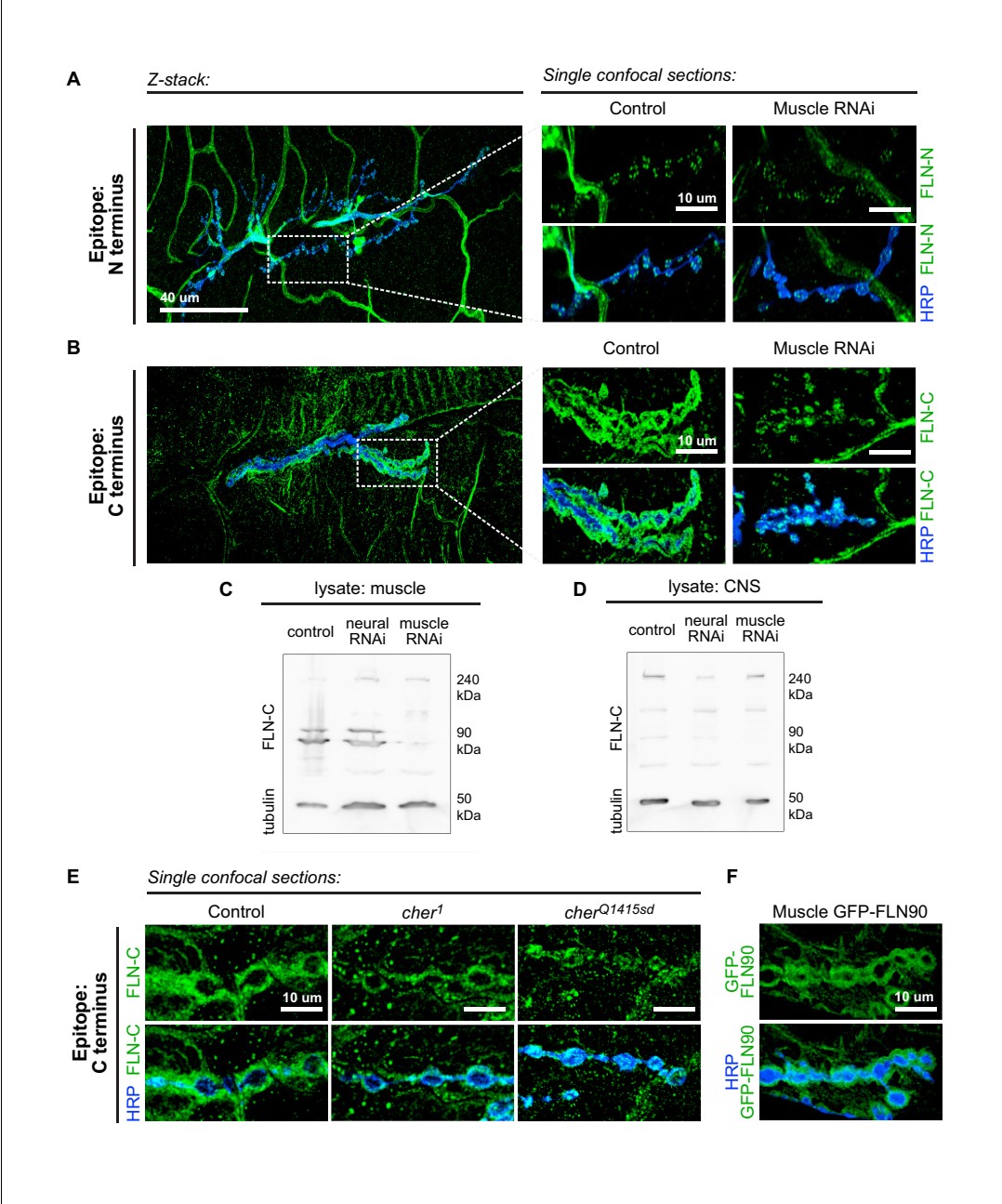

**Figure 4.** The short isoform of filamin localizes to the subsynaptic region. (**A**) Left: a low-magnification, z-stack confocal image of a wildtype NMJ immunostained with FLN-N to detect FLN240 reveals immunostaining in trachea, glia, and nerves. Right: a single 0.4 μm confocal section of individual boutons from the boxed region. (**B**) Images, as in (**A**) of an NMJ immunostained with FLN-C to detect both FLN240 and FLN290. Muscle-specific filamin knockdown (*G14-Gal4/+;UAS-filamin RNAi/+*) abolished the immunoreactivity surrounding the boutons in the control (*G14-Gal4/+;+/+*). (**C** and **D**) Western blots of third-instar larval body walls (**C**) and CNS (**D**) immunoprobed with FLN-C. Muscles were enriched in FLN90, while the CNS preferentially expressed FLN240. Filamin RNAi was driven by G14-Gal4 (muscle) or C155-Gal4 (neural). Tubulin was used as a loading control. (**E**) Single 0.4 μm confocal sections immunostained for both filamin isoforms with FLN-C. Postsynaptic staining was preserved in control (*cher^{Δ12.1}/+*) and cher[1] (*cher^{Δ12.1}/ cher^1*), but not in the *cher^{Q1415sd}* mutant (*cher^{Δ12.1}/cher^{Q1415sd}*). (**F**) Confocal images of NMJs immunostained for GFP-FLN90 expressed in the muscle (*UAS-GFP-FLN90/+;MHCGS/+*). The distribution recapitulates that of endogenous filamin detected by FLN-C. For all images, scale bar: 10 μm.

The following figure supplement is available for figure 4:

**Figure supplement 1.** Filamin expression during NMJ development.

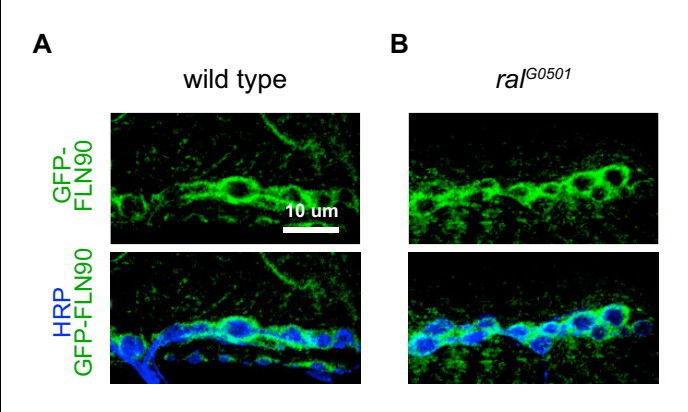

**Figure 5.** FLN90 localizes subsynaptically even in the absence of SSR formation. (**A,B**) Confocal, z-stack images of third-instar NMJs immunostained for GFP-FLN90 expressed in the muscle in (**A**) wild-type (*UAS-GFP-FLN90/+; MHCGS/+*) and (**B**) Ral null (*Ral^G0501^/y; UAS-GFP-FLN90/+;MHCGS/+*) larvae which lack SSR. Scale bars: 10 μm.

filamin expression but filamin localization does not require SSR growth indicates that filamin localization is a cause rather than a consequence of growth of the SSR.

## FLN90 is required for localizing type-A GluRs at the synapse

To determine whether other synaptic features depend on filamin expression, we examined the distribution of glutamate receptors. The fly NMJ expresses two classes of AMPA-type receptors that differ in one of four subunits: type A have the GluRIIA subunit, while type B have GluRIIB (*DiAntonio, 2006*). At a given receptive field opposing each neurotransmitter release site, both types are present, with their relative ratio shifting over the course of development. At immature synapses, type-A receptors predominate, but over time the balance shifts as type-B receptors are added (*Marrus et al., 2004*; *Thomas and Sigrist, 2012*). Knocking down filamin in muscles significantly reduced the synaptic immunoreactivity of GluRIIA without altering the levels of GluRIIA protein in larval lysates (*Figure 6A and B*, and *Figure 6—figure supplement 1A*). FLN90 mediates localization of GluRIIA: synaptic GluRIIA was severely reduced in $cher^{Q1415sd}/cher^{\Delta 12.1}$ but negligibly in $cher^1/cher^{\Delta 12.1}$ (*Figure 6A and B*). In contrast, synaptic GluRIIB levels were unaltered or increased by loss of filamin (*Figure 6C and D*). The change in receptor localization was predominantly a change in the nature rather than the number of receptor clusters. In the absence of filamin, GluRIIA-positive puncta were smaller while GluRIIB-positive puncta were larger (*Figure 6B and D*). The number of GluRIIC-positive puncta, which represent both the type-A and type-B receptors, did not change significantly, but their average size and summed intensity were significantly reduced (*Figure 6E and F*). Neither the average size and number of presynaptic active zone puncta marked by Brp – nor their summed immunofluorescence – changed significantly upon muscle filamin knockdown, although there was a trend to fewer Brp puncta (a 13% decrease, p=0.08). Therefore, the primary consequence of the loss of filamin is a change in the glutamate receptor composition of each receptive field.

To better understand the nature of the shift in GluR composition, we recorded spontaneous and evoked synaptic events using whole-cell recordings in control and filamin-knockdown muscles. The average amplitude of miniature EPSPs did not change in the absence of filamin, but 36% fewer miniature events were detected (*Figure 7A and B*). The amplitude of evoked potentials, judged from peak height or the area under the curve, did not change significantly (*Figure 7C and D*). The selective reduction in mEPSP frequency may be attributed, in part, to the decrease in the number of presynaptic release sites (*Figure 6E and G*), though that decrease was too slight to account for the full effect. It is possible that the reduction in synaptic GluRIIA at a subset of release sites caused some miniature events to fall below the level of detection and into the noise, as has been suggested for GluRIIA-null synapses (*Petersen et al., 1997*).

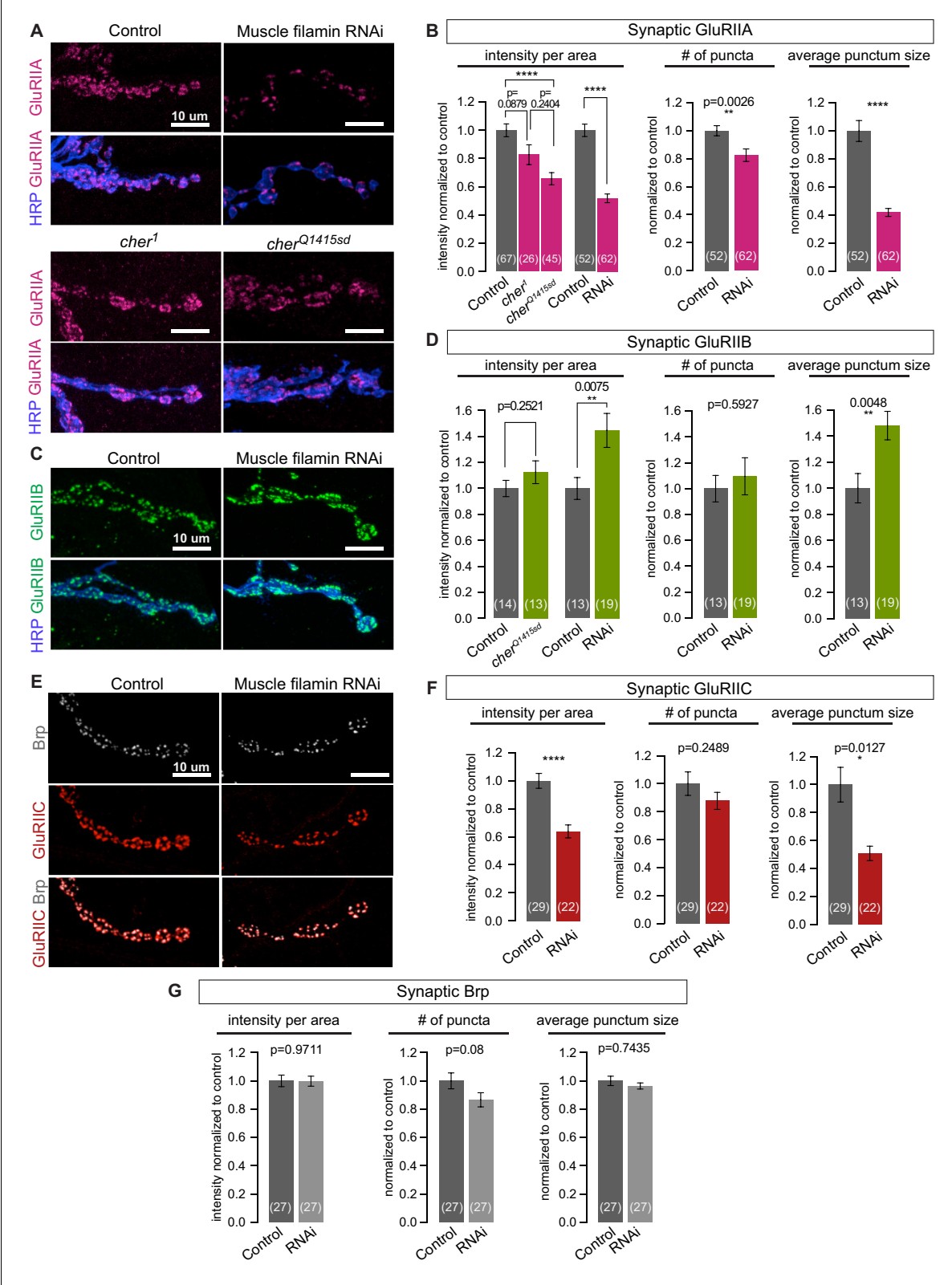

**Figure 6.** Filamin is required for localizing type-A, but not type-B, glutamate receptors. (A,B) Confocal images and quantification of NMJs immunostained for GluRIIA in control larvae (*G14-Gal4/+;+/+*), with muscle-specific knockdown of filamin (*G14-Gal4/+;UAS-filamin RNAi*), and in *cher* mutants (*cher^{Δ12.1}/cher^1* and *cher^{Δ12.1}/cher^{Q1415sd}*). (C,D) Confocal images and quantification of NMJs immunostained for GluRIIB. Genotypes as in (A). (E) Confocal images of NMJs at muscle 6/7, showing the apposition of releases sites (Bruchpilot/Brp immunoreactivity) and GluRIIC, the pan-GluR

*Figure 6 continued on next page*

*Figure 6 continued*

subunit, at NMJs of control (*G14-Gal4/+*) animals versus animals with muscle-specific knockdown against filamin (*G14-Gal4/+;UAS-filamin RNAi*). (F) Quantification of synaptic GluRIIC levels at the indicated genotypes from (E). (G) Quantification of synaptic Brp levels and distribution parameters at the indicated genotypes from (E). Scale bars: 10 um. Number of NMJs quantified is indicated in each bar. Statistical significance was determined with two-tailed unpaired t test except for: (1) in (B), the comparison of intensities of mutant lines used the Kruskal-Wallis test with Dunn's multiple comparisons test; and (2) the Kolmogorov-Smirnov test was used for nonparametric analysis of data with non- normal distributions: in (B), GluRIIA number of puncta and average punctum size; in (F), GluRIIC average punctum size; in (G), Brp average punctum size. ****p<0.0001; other p values indicated on each graph; all error bars indicate ± SEM.

The following figure supplement is available for figure 6:

**Figure supplement 1.** Additional analysis of the effects of filamin loss on GluRs.

Unlike the localization of Syndapin, the localization of GluRIIA was independent of either Ral or SSR formation. At *ral* null synapses, which lack SSR, synaptic GluRIIA distribution was unaltered (***Figure 6—figure supplement 1B***). Thus the localization of GluRIIA occurs downstream of *cher* but independently of *ral* and can occur even in the absence of SSR development.

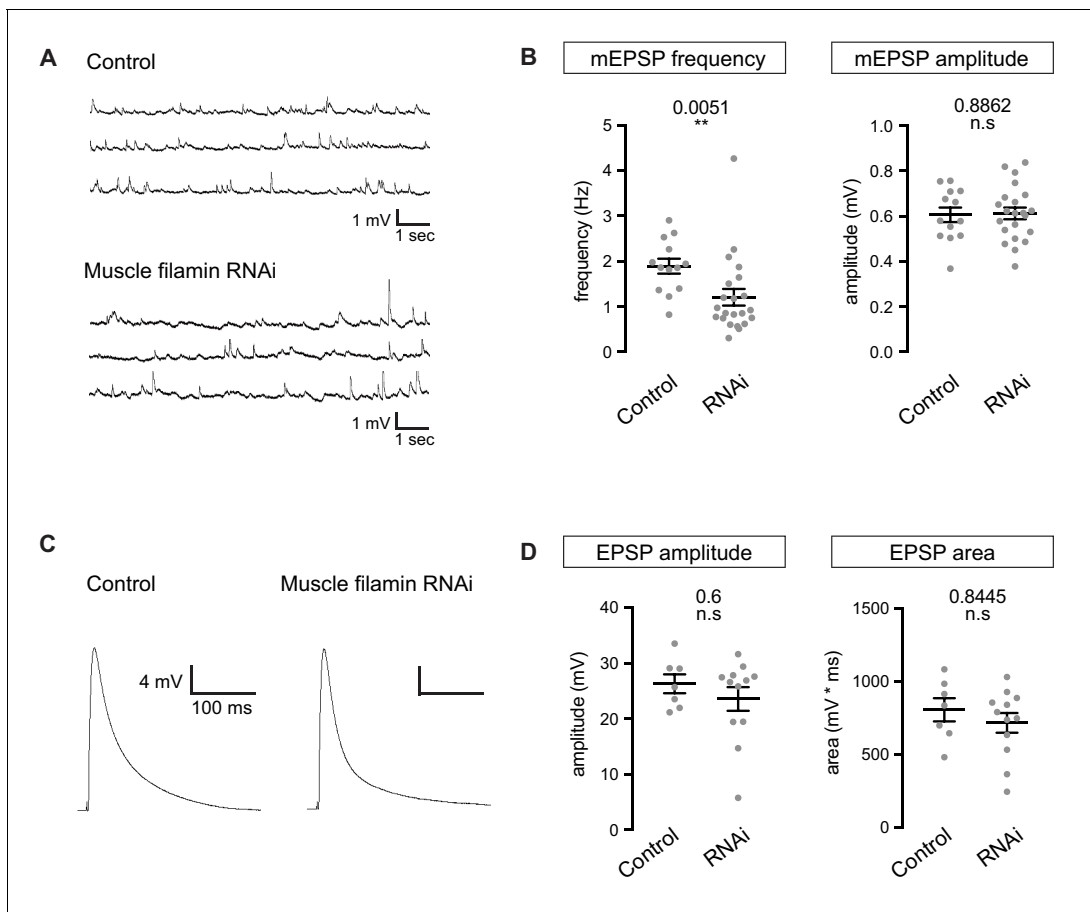

**Figure 7.** Electrophysiological properties of NMJs lacking muscle filamin. (A,B) Representative mEPSP traces from muscles of control larvae (*G14-Gal4/+;+/+*) and larvae with filamin knockdown (*G14-Gal4/+;UAS-filamin RNAi*) and quantification of average mEPSP frequency and amplitude in those genotypes. Y-axis scale bar: 1 mV; x-axis scale bar: 1s. (C,D) Representative EPSP traces from the genotypes in A and quantification of average EPSP amplitude and area under the curve. Y-axis scale bar: 4 mV; x-axis scale bar: 100 ms. Number of NMJs quantified is indicated in each graph. For (B) and (D), statistical significance was determined from Kolmogorov-Smirnov test for nonparametric analysis except for the mEPSP amplitude, whose data showed a normal distribution and two-tailed unpaired t test was used. p values indicated on each graph. All error bars indicate ± SEM.

## Synaptic dPak is required to localize Ral to enable SSR growth

The p21-activated kinase Pak and Ral have been independently reported to regulate the formation of both the SSR (*Parnas et al., 2001*; *Teodoro et al., 2013*) and mammalian dendritic spines (*Penzes et al., 2003*; *Teodoro et al., 2013*; *Zhang et al., 2005*) but have not been placed in a single pathway and downstream effectors of dPak-mediated SSR formation have not been reported. We therefore asked whether Ral and dPak lie in the same pathway with dPak upstream of Ral. The genotype $dPak^6/dPak^{11}$ is functionally null and has a thin SSR (*Hing et al., 1999*; *Parnas et al., 2001*). In $dPak^6/dPak^{11}$ larvae, subsynaptic HA-Ral localization was greatly reduced (*Figure 8A and B*). When we tested the converse using $ral^{G0501}$, a genetically null mutant that survives to the third-instar stage and lacks SSR (*Teodoro et al., 2013*), dPak localization was comparable to that of wild type (*Figure 8C and D*). Thus Ral localization depends on dPak expression and functions genetically downstream of dPak.

We investigated this pathway further by examining another protein required for SSR formation and Pak localization. Pix, a GEF for Rac, localizes Pak to focal complexes and postsynaptically at the fly NMJ and in mammalian spines (*Parnas et al., 2001*; *Zhang et al., 2005*). When $dPix^1$ or $dPix^{P1036}$ are each placed in trans to a dPix deficiency, SSR formation and dPak localization are impaired (*Parnas et al., 2001*); likewise, HA-Ral failed to target the synapse in a $dPix^1/dPix^{P1036}$ background (*Figure 8E and F*). Thus subsynaptic Ral localization, and consequently growth of the SSR, requires both dPak and Pix.

## dPak localizes Ral to the synapse via its kinase activity

Pak is a kinase which, depending on the context, may also function as a kinase-independent scaffold (*Bokoch, 2003*; *Daniels et al., 1998*; *Frost et al., 1998*). At the fly NMJ, kinase activity is required for recruitment of DLG and GluRIIA (*Albin and Davis, 2004*). Having observed that synaptic dPak regulates Ral localization, we tested whether this is also mediated by its kinase activity by taking advantage of existing mutant alleles. The $dPak^3$ allele is a G-to-D point mutation in the DFG catalytic triplet conserved in all kinases and is thought to render the kinase domain inactive (*Hing et al., 1999*); the $dPak^{21}$ allele produces a truncated dPak missing the entire kinase domain (*Newsome et al., 2000*). When each was placed in trans to a $dPak^{11}$ (null) allele, synaptic recruitment of epitope-tagged Ral was severely diminished (*Figure 8A and B*). Therefore, the kinase activity of dPak is necessary for recruitment of Ral to the synapse.

## Filamin is required for synaptic localization of dPak

The placement of dPak upstream of Ral in the pathway for SSR development raised the question of dPak's relationship to filamin. Indeed, the reported NMJ phenotypes for dPak (*Albin and Davis, 2004*; *Parnas et al., 2001*) closely resemble those we found for filamin: a selective decrease in synaptic GluRIIA, a decrease in synaptic DLG, and a lack of SSR. Moreover, mammalian Pak binds and phosphorylates FLNA (*Vadlamudi et al., 2002*). We therefore examined their relationship at the fly NMJ. In muscles lacking filamin, either due to muscle-specific knockdown of filamin or the $cher^{Q1415sd}/cher^{\Delta 12.1}$ genotype, synaptic levels of both endogenous and GFP-tagged dPak were significantly reduced (*Figure 9*). Indeed, GFP-dPak was almost undetectable at synapses when filamin was knocked down (*Figure 9—figure supplement 1A*). Conversely, when dPak was knocked down in muscle, filamin was still present at the synapse, although reduced relative to control NMJs (*Figure 9C and D*). Thus filamin and dPak lie in a pathway with filamin most likely acting upstream of dPak localization. To determine whether *Drosophila* FLN90 and dPak also interact physically, we employed in situ proximity ligation assay (PLA), which detects when two proteins are within 40 nm of each other. Because the postsynaptic compartment is crowded and therefore may yield PLA positive signal without a direct interaction of the two proteins, we instead expressed FLN90 and dPak in a heterologous system. V5-dPak was coexpressed in HEK293T cells with either GFP-FLN90 or GFP alone as a control and their association was tested with anti-V5 and anti-GFP antibodies. PLA-positive cells represented a much higher fraction of GFP+ cells when V5-dPak was co-expressed with GFP-FLN90 (484 of 973) than with GFP alone (142 of 979) (*Figure 9G*). Therefore, the filamin-Pak interaction is conserved across species and suggests that synaptic dPak localization is achieved through its direct binding to filamin.

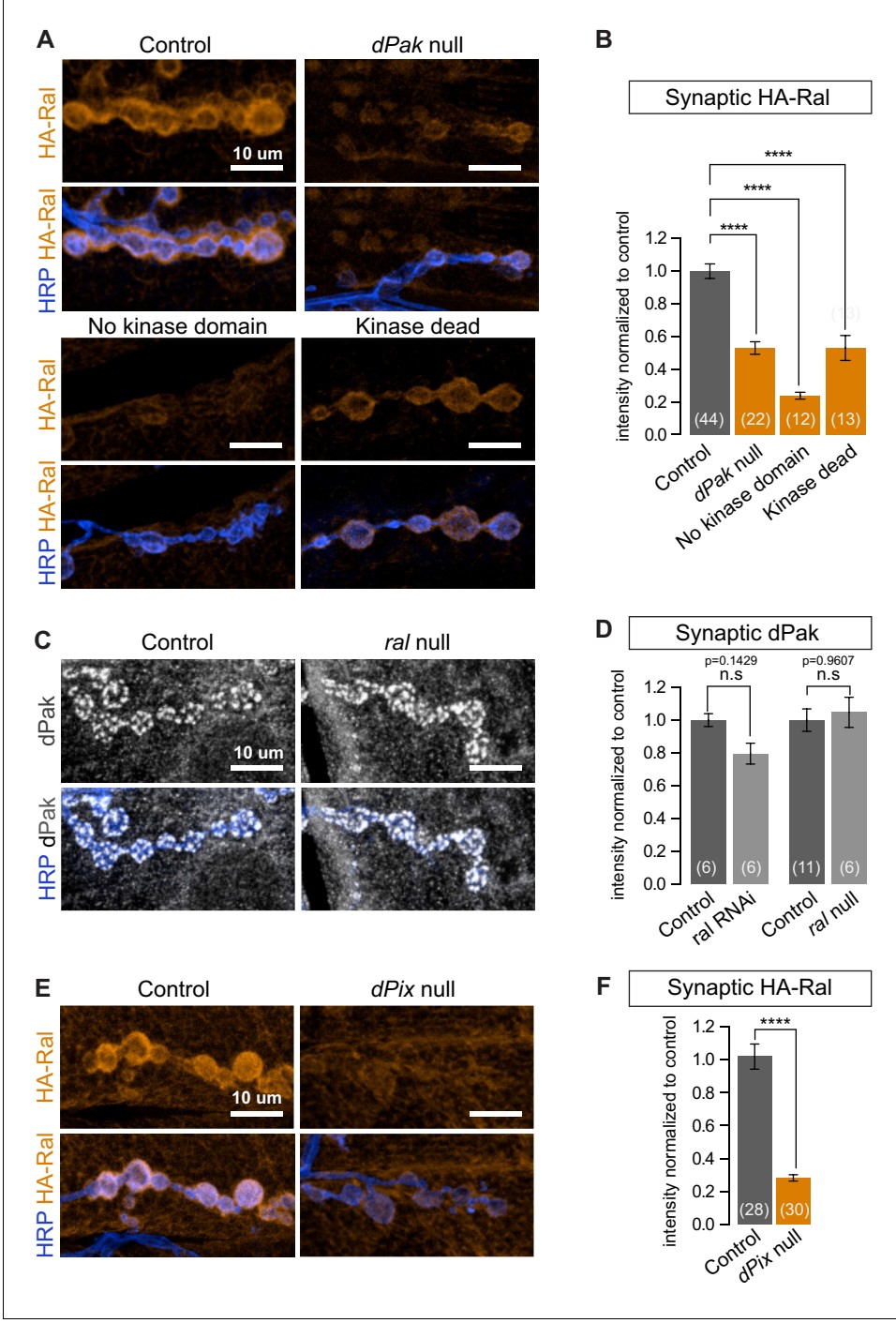

**Figure 8.** dPak and its kinase activity are required for localizing Ral. (**A,B**) Confocal images and quantification of subsynaptic HA-Ral at NMJ of the genotypes: control (*UAS-HA-Ral/G14-Gal4;dPak$^{11}$/+*), dPak null (*UAS-HA-Ral/G14-Gal4;dPak$^{11}$/dPak$^{6}$*), no kinase domain (*UAS-HA-Ral/G14-Gal4; dPak$^{11}$/dPak$^{21}$*), and kinase dead (*UAS-HA-Ral/G14-Gal4; dPak$^{11}$/dpak$^{3}$*). (**C,D**) Confocal images and quantification of dPak immunoreactivity in control (*ral$^{G0501}$/+*) and ral null larvae (*ral$^{G0501}$/y*) and (in D) also upon knockdown of Ral (*G14-Gal4/+;+/+ and G14-Gal4/+;UAS-ral RNAi*). (**E,F**) Confocal images and quantification of subsynaptic HA-Ral at NMJs of control (*dPix$^{P1036}$/+;UAS-HA-Ral/MHCGS*) and dPix null larvae (*dPix$^{P1036}$/dPix$^{1}$;UAS-HA-Ral/MHCGS*). Scale bars: 10 μm. Number of NMJs quantified is indicated in each graph. For (**D**), statistical significance determined with Kolmogorov-Smirnov test; for (**F**), two-tailed unpaired t test; and for (**B**), multiple comparisons were performed using Kruskal-Wallis test with Dunn's multiple comparisons test. ****p<0.0001; other p values are indicated on each graph; 'n.s' = not significant. All error bars indicate ± SEM.

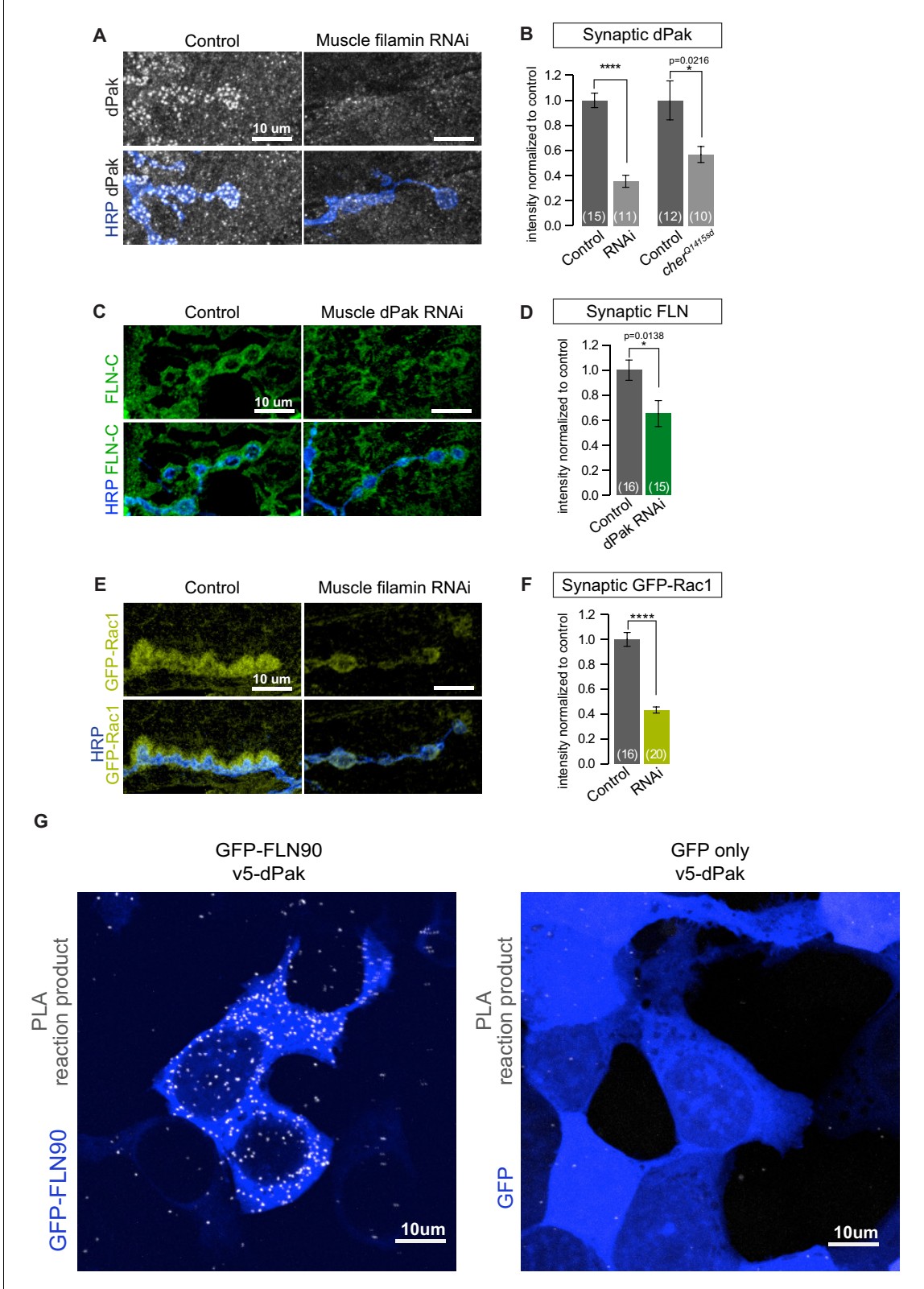

**Figure 9.** Filamin is required for localizing dPak and its activator Rac1. (**A,B**) Confocal images and quantification of dPak immunoreactivity in control NMJs (*G14-Gal4/+;+/+*) and with muscle-specific knockdown of filamin (*G14-Gal4/+;UAS-filamin RNAi*). (**C,D**) Representative single 0.4 μm confocal sections and quantification of FLN-C immunoreactivity at control NMJs (*G14-Gal4/+;+/+*) and with muscle-specific knockdown of dPak (*G14-Gal4/+; UAS-dPak RNAi/+*). (**E,F**) Confocal images and quantification of NMJs immunostained for GFP-Rac1 expressed under control by the endogenous Rac1

*Figure 9 continued on next page*

*Figure 9 continued*

promoter in control larvae (*G14-Gal4/Rac::GFP-Rac;+/+*) and with muscle-specific knockdown of filamin (*G14-Gal4/Rac::GFP-Rac;UAS-filamin RNAi*). Scale bars: 10 um. Number of NMJs quantified is indicated in each graph. Statistical significance determined with two-tailed unpaired t test. ****$p<0.0001$; other p values are indicated on each graph. All error bars indicate ± SEM. (**G**) In-situ PLA (Proximity Ligation Assay) detection of *Drosophila* GFP-FLN90 and V5-dPak (left) and GFP and V5-dPak (right, as negative control) expressed in HEK293T cells. Immunoreactivities are shown for GFP (blue) to identify transfected cells and PLA-positive signal (gray).

The following figure supplement is available for figure 9:

**Figure supplement 1.** Loss of filamin disrupts synaptic targeting of GFP-dPak.

Across systems, Pak is usually found in a signaling complex with its direct activators such as Rac, and studies have suggested that this role for Rac is conserved at the fly NMJ (*Albin and Davis, 2004*). Since mammalian Pak and Rac are both reported to bind FLNA (*Ohta et al., 1999*; *Vadlamudi et al., 2002*), we tested whether synaptic Rac localization is also filamin-dependent. GFP-tagged Rac1 expressed under its endogenous promoter (*Abreu-Blanco et al., 2014*) was present both pre- and postsynaptically at the NMJ. However, with filamin knockdown in muscle, the postsynaptic signal was barely detectable, causing a significant reduction in overall synaptic levels (combined pre-and-postsynaptic immunoreactivity; *Figure 9E and F*). Together, the genetics indicate that filamin is necessary for postsynaptic localization of dPak and its signaling partner Rac.

## Discussion

In this study we demonstrate that filamin is essential to orchestrating the recruitment of core components of the postsynaptic machinery. Filamin is a highly conserved protein whose loss of function is associated with neurodevelopmental disorders. In humans, mutations in the X-linked FLNA cause periventricular heterotopia, a disorder of cortical malformation with a wide range of clinical manifestations such as epilepsy and neuropsychiatric disturbances (*Feng and Walsh, 2004*; *Fox et al., 1998*; *Robertson, 2005*). Studies in rodent models have shown that abnormal filamin expression causes dendritic arborization defects in a TSC mouse (*Zhang et al., 2014*) and that filamin influences neuronal proliferation (*Lian et al., 2012*). Filamin is present in acetylcholine receptor clusters at the mammalian NMJ (*Bloch and Hall, 1983*; *Shadiack and Nitkin, 1991*), but its function there is unknown. In lysates of the mammalian brain, filamin associates with known synaptic proteins such as Shank3, Neuroligin 3, and Kv4.2 (*Petrecca et al., 2000*; *Sakai et al., 2011*; *Shen et al., 2015*). A recent report indicated that filamin degradation promotes a transition from immature filopodia to mature dendritic spines (*Segura et al., 2016*), a phenomenon that is likely to be related to the actin-bundling properties of the long isoform of filamin. Data in the present study have uncovered a novel pathway that does not require the actin-binding domain of filamin. In this pathway, postsynaptically localized filamin, via Pak, directs two distinct effector modules governing synapse development and plasticity: (1) the Ral-exocyst pathway for activity-dependent membrane addition and (2) the composition of glutamate receptor clusters (*Figure 10*). These pathways determine key structural and physiological properties of the postsynapse.

Although loss of filamin had diverse effects on synapse assembly, they were selective. Muscle-specific knockdown or the *cher*^{Q1415sd} allele disrupted type-A but not type-B GluR localization at the postsynaptic density (*Figure 6*). Likewise, the phenotypes for muscle filamin were confined to the postsynaptic side: the presynaptic active zone protein Brp and overall architecture of the nerve endings were not altered by muscle-specific knockdown (*Figure 1—figure supplement 1C*, *Figure 6E and G*). The specificity of its effect on particular synaptic proteins, and the absence of the actin-binding region in FLN90, suggests that filamin's major mode of action here is not overall cytoskeletal organization, but rather to serve as a scaffold for particular protein-protein interactions.

### Filamin is a platform for postsynaptic structural maturation and plasticity

Analysis of the distribution of the SSR marker Syndapin and direct examination of the subsynaptic membrane by electron microscopy revealed that formation of the SSR required filamin. Genetic

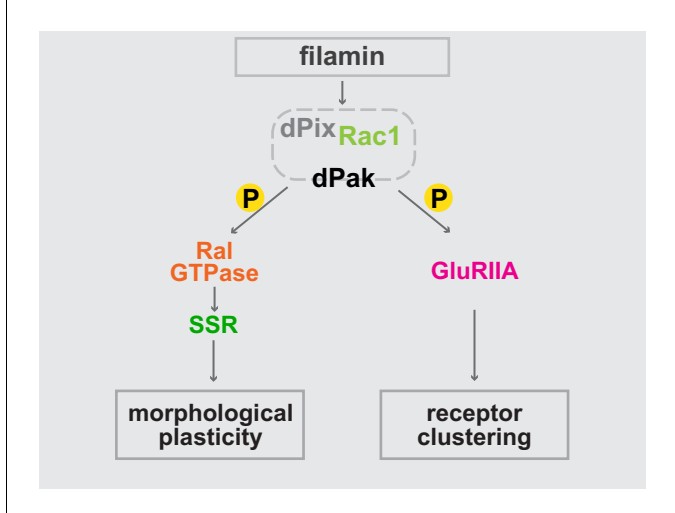

**Figure 10.** Bifurcated filamin-dependent pathways for SSR growth and receptor localization. The epistatic relationships of proteins required for SSR growth and GluRIIA localization at the synapses are diagrammed. Filamin is required to localize dPak to the synapse and dPak is required for Ral localization, without which exocyst- and activity-dependent growth of the SSR cannot occur. Independently of Ral, dPak localization is also needed for GluRIIA to cluster at the postsynaptic density which is otherwise composed only of type-B receptors. In both branches of the pathway, dPak acts as a kinase and with its associated partners Pix and Rac1.

analysis uncovered a sequential pathway for SSR formation from filamin to the Pak/Pix/Rac signaling complex, to Ral, to the exocyst complex and consequent membrane addition. The SSR is formed during the second half of larval life and may be an adaptation for the low input resistance of third-instar muscles. Like dendritic spines, the infoldings of the SSR create biochemically isolated compartments in the vicinity of postsynaptic receptors and may shape physiological responses, although first-order properties of the synapse, such as mini- or EPSP amplitude are little altered in mutants that lack an SSR (*Gorczyca et al., 2007*; *Kumar et al., 2009b*). The formation of the SSR requires transcriptional changes driven by Wnt signaling and nuclear import (*Korkut and Budnik, 2009*; *Mosca and Schwarz, 2010*; *Packard et al., 2002*), proteins that induce membrane curvature (such as Syndapin, Amphiphysin, and Past1) (*Koles et al., 2015*; *Kumar et al., 2009a*), and Ral-driven, exocyst-dependent membrane addition (*Teodoro et al., 2013*). The activation of Ral by Ca$^{2+}$ influx during synaptic transmission allows the SSR to grow in an activity-dependent fashion. The localization of Ral to the region surrounding the bouton appears crucial to determining the site of membrane addition because Ral localization precedes SSR formation and exocyst recruitment and because exocyst recruitment occurred selectively surrounding boutons even when Ca$^{2+}$-influx occurred globally in response to calcimycin (*Teodoro et al., 2013*). We have now shown that Ral localization, and consequently exocyst recruitment, membrane growth, and the presence of the SSR marker Syndapin, are all dependent on a local action of filamin at the synapse. FLN90, the filamin short isoform, localized to sites of synaptic contact and indeed surrounded the boutons just as does Ral (*Figure 4*). When this postsynaptic filamin was removed by muscle-specific filamin knockdown or the *cher*$^{Q1415sd}$ allele, the downstream elements of the pathway, Pak, Rac, Ral, the exocyst, and Syndapin, were no longer synaptically targeted. The mislocalization is not a secondary effect of loss of the SSR but likely a direct consequence of filamin loss, as Pak and Ral can localize subsynaptically even in the absence of the SSR (*Figure 6—figure supplement 1B*, *Teodoro et al., 2013*). Unlike the likely mode of action of nuclear signaling by Wnt, the delocalization of Ral was not a consequence of altered protein production; its expression levels did not change (*Figure 1—figure supplement 1A*). Thus filamin may be viewed as orchestrating the formation of the SSR and directing it to the region surrounding synaptic boutons.

## Filamin regulates postsynaptic receptor composition

The second major feature of the filamin phenotype was the large reduction in the levels of the GluR-IIA receptor subunit from the postsynaptic membranes. GluRIIA and GluRIIB differ in their electrophysiological properties and subsynaptic distribution (*DiAntonio et al., 1999*; *Marrus et al., 2004*). Because type B GluRs, which contain the IIB subunit, desensitize more rapidly than type A, the relative abundance of type A and type B GluRs is a key determinant of postsynaptic responses and changes with synapse maturation. The selective decrease in GluRIIA at filamin-null NMJs is likely a consequence of dPak mislocalization: filamin-null NMJs lack synaptic dPak, and dPak null NMJs lack synaptic GluRIIA (*Albin and Davis, 2004*; *Parnas et al., 2001*). Moreover, the first-order electrophysiological properties at NMJs lacking filamin resembled those reported at NMJs missing dPak (*Parnas et al., 2001*). In our study, though, only the change in mEPSP frequency was statistically significant. At filamin-null NMJs, the decrease in GluRIIA is accompanied by an increase in GluRIIB, suggestive of a partial compensation that could account for the relatively normal synaptic transmission. Because the IIA and IIB subunits differ in desensitization kinetics and regulation by second messengers (*DiAntonio, 2006*), functional consequences of filamin loss may become more apparent with more extensive physiological characterizations at longer time scales.

While both SSR growth and receptor composition required the kinase activity of dPak, receptor composition was independent of Ral and thus represents a distinct branch of the pathway downstream of dPak. As with Ral, the loss of GluRIIA from the synapse was due to delocalization and not a change in expression of the protein, consistent with unaltered GluRIIA transcripts in dPak null animals (*Albin and Davis, 2004*). Thus filamin, via dPak, alters proteins with functional significance for the synapse as well as its structural maturation.

## Filamin as a postsynaptic scaffold

Mammalian filamin, via its many Ig-like repeats, has known scaffold functions in submembrane cellular compartments (*Popowicz et al., 2006*; *Stossel et al., 2001*) and filamin is therefore likely also to serve as a scaffold at the fly NMJ. Our epistasis data indicate that filamin recruits Ral through recruitment of a signaling complex already known to function at the fly NMJ: dPak and its partners dPix and Rac (*Albin and Davis, 2004*; *Parnas et al., 2001*). Mammalian filamin is reported to directly interact with Ral during filopodia formation (*Ohta et al., 1999*), however the details of their interaction at the fly NMJ are less clear. Because Ral localization requires filamin to recruit dPak and dPix and specifically requires the kinase activity of dPak, it is possible that either Ral or filamin need to be phosphorylated by dPak to bind one another. Mammalian filamin interacts with some components of the Pak signaling complex (*Bellanger et al., 2000*; *Ohta et al., 1999*) and is a substrate of Pak (*Vadlamudi et al., 2002*). We have now shown that *Drosophila* filamin and PAK interact when coexpressed in HEK cells, and thus a direct scaffolding role for FLN90 in the recruitment of Pak and the organization of the postsynapse is likely.

The overlapping but different distributions of filamin and its downstream targets indicate that its scaffolding functions must undergo regulation by additional factors. The proteins discussed here take on either of two patterns at the synapse. Some, like Ral, Syndapin, and filamin itself, are what we have termed subsynaptic and, like the SSR, envelope the entire synaptic bouton. Others, like dPak and its partners and the GluRIIA proteins, are concentrated in much smaller regions, immediately opposite presynaptic active zones, where the postsynaptic density (PSD) is located. We hypothesize that filamin interacts with additional proteins, including potentially transsynaptic adhesion proteins, that localize filamin to the subsynaptic region and also govern to which of the downstream effectors it will bind. Indeed, it appears paradoxical that dPak, though predominantly at the postsynaptic density is nonetheless required for Ral localization throughout the subsynaptic region. If dPak is needed to phosphorylate either filamin or Ral to permit Ral localization, the phosphorylations outside the PSD may be due to low levels of the dPak complex in that region; synaptic dPak was previously shown to be a relatively mobile component of the PSD (*Rasse et al., 2005*).

## Evidence for FLN90-specific function at the synapse

Filamin was the first nonmuscle actin-crosslinking protein to be discovered (*Hartwig and Stossel, 1975*; *Stossel and Hartwig, 1975*). With an actin-binding domain at the N terminus, the long isoform of filamin and its capacity to integrate cellular signals with cytoskeletal dynamics have

subsequently been the focus of the majority of the filamin literature (*Nakamura et al., 2011*; *Popowicz et al., 2006*; *Stossel et al., 2001*). At the NMJ, however, this was not the case. Several lines of evidence indicated that the short FLN90 isoform of filamin, which lacks the actin-binding domain, plays an essential role in postsynaptic assembly. First, the short FLN90 isoform was the predominant and perhaps the only isoform of filamin found expressed in the muscles. Second, both endogenous and overexpressed FLN90 localized subsynaptically. Third, loss of the short isoform disrupted localization of postsynaptic components while lack of just the long isoform had little or no effect. Lastly, exogenous expression of just the short isoform in filamin null background sufficiently rescued the defect in SSR growth. The modest postsynaptic phenotypes of the *cher$^1$*allele, which predominantly disrupts the long isoform, may be due to small effects of the allele on expression of the short isoform or may be an indirect consequence of the presence of the long isoform in the nerve terminals.

The existence of the short isoform has been reported in both flies and mammals and may be produced either by transcriptional regulation or calpain-mediated cleavage (*Browne et al., 2000*; *van der Flier and Sonnenberg, 2001*; *Gorlin et al., 1990*; *Savoy and Ghosh, 2013*; *Wang et al., 2007*). The short isoform can be a transcriptional co-activator (*Loy et al., 2003*; *Wang et al., 2007*), but its functional significance and mechanisms of action have been largely elusive. The short isoform has little or no affinity for actin (*Nakamura et al., 2007*), but most of the known sites for other protein-protein interactions are shared by both isoforms. Thus the structure of FLN90, with nine predicted Ig repeats and likely protein-protein interactions, is consistent with a scaffolding function to localize key synaptic molecules independent of interactions with the actin cytoskeleton.

Our study has introduced filamin as a major contributor to synapse development and organization. The severity of the phenotypes indicates filamin has a crucial role that is not redundant with other scaffolding proteins. The effects of filamin encompass several much-studied aspects of the *Drosophila* NMJ: the clustering and subunit subtype of glutamate receptors and the plastic assembly of specialized postsynaptic membrane structures. The pathways that govern these two phenomena diverge downstream of Pak kinase activity and are dependent on filamin for the proper localization of key signaling modules in the pathways. By likely acting as a scaffold protein, the short isoform of filamin may function as a link between cell surface proteins, as yet unidentified, and postsynaptic proteins with essential localizations to and functions at the synapse. Because many of the components of these pathways at the fly NMJ are also present at mammalian synapses and can interact with mammalian filamin, a parallel set of functions in CNS dendrites merits investigation.

## Materials and methods

### *Drosophila* husbandry and genetics

Flies were maintained on standard medium at 25°C. For larval collection, eggs were laid and grown on grape juice plates and yeast paste at 25°C. For RNAi experiments, collection of RNAi expressing strains and their controls were set up at 29°C to maximize the efficiency of knockdown. GFP-expressing balancer chromosomes were used to facilitate genotyping of larvae.

For tissue-specific transgene expression, *G14-Gal4* (*Aberle et al., 2002*) and *MHCGS* (without addition of RU486) (*Osterwalder et al., 2001*) were used to drive UAS constructs' expression in muscles and *C155-Gal4* for expression in neurons. Transgenic UAS lines used, UAS-driven RNAi lines, and mutant alleles used are described in tables below:

### Gal4 drivers used

| Driver name | Description | Reference |
| --- | --- | --- |
| G14-Gal4 | All somatic muscles, early expression; on chromosome 2 | (*Aberle et al., 2002*) |
| MHC-Geneswitch (MHCGS) | Chemically-inducible variant on the MHC driver. Used in this and in *Teodoro et al. (2013)* without chemical induction by RU-486, as its 'leakiness' inpost-embryonic stages is sufficient to allow expression of transgenic HA-Ral and GFP-FLN90; on chromosome 3 | (*Osterwalder et al., 2001*) |

## UAS transgenic lines used

| Name | Description | Reference |
|------|-------------|-----------|
| UAS-Ral$^{GTP}$ | Constitutively active Ral (G20V) on chromosome 3 | (*Mirey et al., 2003*) |
| UAS-Ral$^{GDP}$ | Constitutively inactive Ral (S25N) on chromosome 2 | (*Mirey et al., 2003*) |
| UAS-HA-Ral | Wild type Ral with an N terminus HA tag; on chromosome 2 or 3 | This study |
| UAS-Cher RNAi$^{HMS}$ | RNAi against filamin; on chromosome 3 | Bloomington stock center |
| UAS-GFP-FLN90 | FLN90 isoform with an N terminus EGFP tag; on chromosome 2 | This study |
| UAS-dPak RNAi$^{HM}$ | RNAi against dPak; on chromosome 3 | Bloomington stock center |

## Mutant lines used

| Genotype | Description | Reference / Source if otherwise |
|----------|-------------|----------------------------------|
| cher$^{Q1415sd}$ | Also referred to as filamin$^{sko}$ (*Li et al., 1999*); EMS-induced mutant; functionally null allele | (*Li et al., 1999*; *Sokol and Cooley, 1999*) |
| cher$^{Δ12.1}$ | Deficiency covering the entire *cheerio* transcription unit | (*Li et al., 1999*) |
| cher$^1$ | P-element disruption of *cheerio* locus | (*Robinson et al., 1997*) |
| dPak$^6$ | Stop codon in CRIB domain; genetic null | (*Hing et al., 1999*) / Bloomington Stock Center |
| dPak$^{11}$ | Stop codon in the middle; genetic null | (*Hing et al., 1999*) / Bloomington Stock Center |
| dPak$^{21}$ | Q382stop mutation generating truncated dPak lacking kinase domain | (*Newsome et al., 2000*) / N. Harden Lab |
| dPak$^3$ | G569D mutation in kinase domain rendering it kinase dead | (*Hing et al., 1999*) / N. Harden Lab |
| dPix$^1$ | EMS-induced mutation; genetic null | (*Parnas et al., 2001*)/ M. Pecot Lab |
| dPix$^{P1036}$ | P-element-mobilized excision; genetic null | (*Parnas et al., 2001*)/ M. Pecot Lab |
| ral$^{G0501}$ | Genetic null | Bloomington Stock Center |

## Recombinants and other lines used

| Label | Genotype / description | Reference(s) |
|-------|------------------------|--------------|
| Muscle Ral$^{GTP}$ | UAS-Ral$^{GTP}$, MHC-Geneswitch | (*Osterwalder et al., 2001*) (MHC-Geneswitch); (*Mirey et al., 2003*) (UAS-Ral$^{GTP}$); (*Teodoro et al., 2013*) (Recombinant) |
| GFP-Rac1 | Rac1::GFP-Rac1; expresses GFP-tagged Rac1 under Rac1 regulatory sequence. | (*Abreu-Blanco et al., 2014*) |

## Generation of transgenic flies

For UAS-HA-Ral and UAS-GFP-FLN90, cDNA construct LD21679 was cloned into the vector pTHW and RE44980 into PTGW (Drosophila Genomics Resource Center, Bloomington, IN) using the Gateway system and incorporated using P element transformation (BestGene, Chino Hills, CA).

## Cloning

All cloning for this study was performed using the Gateway Cloning System (ThermoFisher Scientific, Waltham, MA) with its respective reagents. PCR for subcloning was performed using the Expand High Fidelity PCR System (Roche, Basel, Switzerland).

For UAS-HA-Ral, the cDNA sequence was subcloned from LD21679 obtained from the Drosophila Genomics Resource Center (Bloomington, IN) into the pDONR221 entry vector. The attB primer sequences used for this are:

attB1:

5'– GGGGACAAGTTTGTACAAAAAAGCAGGCTTCATGAGCAAGAAGCCGACAGCC -–3'

attB2:

5' – GGGGACCACTTTGTACAAGAAAGCTGGGTCTAAAGTAGGGTACACTTAAGTC – 3'

The destination vector (pTHW) is a pUAST vector containing an HA-tag at the N terminus, obtained from the Drosophila Genomics Resource Center (Bloomington, IN).

For UAS-GFP-FLN90, the cDNA sequence was subcloned from RE44980 obtained from the Drosophila Genomics Resource Center (Bloomington, IN) into the pDONR221 entry vector. The attB primer sequences used for this are:

attB1:

5'-GGGGACAAGTTTGTACAAAAAAGCAGGCTTCATGCCTAGCGGTAAAGTAGAC – 3'

attB2:

5'-GGGGACCACTTTGTACAAGAAAGCTGGGTCCTACACATCGATCTGGAATGG – 3'

The destination vector (pTGW) is a pUAST vector containing a GFP-tag at the N terminus, obtained from the Drosophila Genomics Resource Center (Bloomington, IN).

Expression of respective proteins were verified in *Drosophila* S2 cells co-expressing a construct for Actin-Gal4, with immunofluorescence and biochemistry.

## P-element transformation

The verified expression vectors were sent to BestGene (Chino Hills, CA) for injection into embryos with w1118 background and incorporation via p-element transformation.

## Immunohistochemistry

Third instar larvae were pinned down onto Sylgard-coated plates using 0.1 mm minutien pins (Fine Science Tools, Foster City, CA) and dissected in PBS (phosphate-buffered saline) using techniques similar to those described in *Brent et al., 2009*. Gut and fat body were removed, while the CNS was kept intact until after fixation. Larval fillets were fixed with either PFA or Bouin's Fixative (see below), then extensively washed using PBT (0.3% Triton-X (Sigma-Aldrich, St. Louis, MO) in PBS). Blocking was done for 30 min–1 hr at room temperature, using 5% normal goat serum in PBT. Primary antibody incubation was performed overnight in 4°C, in blocking solution. Subsequently, larvae were extensively washed using PBT and secondary antibodies were added in blocking solution for 1 hr at room temperature. After extensive washing using PBT, larvae were mounted onto slides in VectaShield (Vector Labs, Burlingame, CA) as the mounting medium, covered with coverslips which were then sealed using nail polish (Sally Hansen Hard As Nails Xtreme Wear). Slides were stored in −20°C until ready to image.

## Fixative conditions

- PFA: 4% Paraformaldehyde diluted using 1X PBS from a 37% stock (Electron Microscopy Services, Hatfield, PA); 20 min at room temperature
- · Bouin's: Bouin's Fixative Solution (Ricca Chemical Company, Arlington, TX); 5 min at room temperature

## Primary antibodies used in this study

| Antigen and clone ID | Species | Concentration | Fixative | Reference / Source |
|---|---|---|---|---|
| Brp (NC82) | Mouse | 1:100 | Bouin's | DSHB (Developmental Studies Hybridoma Bank, Iowa City, IA) |
| Dlg (4 F3) | Mouse | 1:500 | PFA | DSHB |
| dPak | Rabbit | 1:2000 | PFA | (*Harden et al., 1996*) / N. Harden Lab |
| Filamin (C terminus) (43-D) | Rabbit | 1:100 | Bouin's | (*Li et al., 1999*) / T. Hays Lab |

| | | | | |
|---|---|---|---|---|
| Filamin (N terminus/aa 189–482) | Rat | 1:800 | Bouin's | (*Külshammer and Uhlirova, 2013*) / M. Uhlirova Lab |
| GFP | Rabbit | 1:1000 | For GFP-FLN90, Bouin's; for GFP-dPak and GFP-Rac1, PFA | Life technologies / Molecular Probes / RRID:AB_221569 |
| GluRIIA, supernatant | Mouse | 1:50 | Bouin's | DSHB |
| GluRIIB | Rabbit | 1:2500 | PFA | (*Marrus et al., 2004*) / A. DiAntonio Lab |
| HA (3 F10) | Rat | 1:200 | PFA | Roche / RRID:AB_2314622 |
| Ral | Guinea pig | 1:800 | PFA | (*Teodoro et al., 2013*) |
| Sec5 (22 A2) | Mouse | 1:35 | PFA | (*Murthy et al., 2003*) |
| Syndapin | Guinea pig | 1:1000 | PFA | Gift from M. Ramaswami |

## Confocal imaging and data analysis

Muscle 6/7 from segments A2 and A3 were imaged and analyzed, unless otherwise stated. All confocal images were acquired using the LSM700 and LSM710 confocal microscopes (Zeiss). For all images, the pinhole size was set to one airy unit, making each optical section 0.4 um.

### Analysis of immunofluorescence intensity

For analyzing immunofluorescence intensity, maximum intensity projections from z-stacks were used. For consistency, quantification was semi-automated using a macro on ImageJ (NIH) (See below for details). Briefly, the intensity was measured and divided by the area of its respective ROI and the background intensity/area was subtracted. To define an ROI for measurements of synaptic immunofluorescence for a given protein, the area of the presynaptic endings was first selected on the basis of anti-HRP staining. At a control synapse, this area was then expanded by a constant and empirically determined distance so as to encompass sufficient of the surrounding muscle to represent the maximum subsynaptic area occupied by that protein. That distance was then used to define the ROI in all genotypes for that protein. For different proteins, the values ranged from 0.1–0.9 μm. The background intensity for muscle cytoplasm in a given specimen was obtained by expanding the anti-HRP area by a 2 μm radius (well beyond the subsynaptic region) and then defining an ROI as a shell 5.3-um-thick surrounding that zone. All quantifications are shown as mean values normalized to respective controls, with standard errors. For pairwise comparisons, statistical significance was determined using a two-tailed unpaired t test with Welch's correction for unequal SDs, unless otherwise indicated in figure legends. All multiple comparisons in were performed using Kruskal-Wallis test with Dunn's multiple comparisons test.

The macro used to quantify synaptic immunofluorescence is summarized as follows:

1. ) Z-projection
   - Maximum intensity projection
2. ) Segmentation of ROI
   - Based on the area delimited by the presynaptic HRP staining
   - Threshold, then binarize, the HRP channel using ImageJ's automatic thresholding algorithm → 'HRP-ROI'
   - Expand the HRP-ROI by a constant specific to the protein analyzed. For example, for Syndapin the ROI was expanded by 0.9 um to appropriately capture all synaptic Syndapin even at maximal expression.
3. Measurement of synaptic intensity
   - Intensity measurement limited to the ROI set above
4. Background measurement
   - Delineate a 5.3-um-thick ROI/annulus that is 2 um radius away from HRP-ROI
   - Intensity measurement limited to the background ROI
5. Synaptic intensity (a.u / area) subtracted by background intensity (a.u / area)

## Analysis of puncta number and size

For analyzing the number and size of Brp/GluR –positive puncta at the synapse (*Figure 5*), maximum intensity projections from z-stacks were used. For consistency, quantification was semi-automated using a macro on ImageJ (NIH), summarized as follows:

1. Z-projection
   - Maximum intensity projection
2. Eliminate background and nonsynaptic Brp/GluR signals
   - Eliminate signals ~1 um beyond the region covered by the HRP-positive signal (i.e., nerve terminal) safely presumed not to be synaptic
3. Segmentation of Brp/GluR puncta
   - Threshold pixels with intensities between a 255 (maximum) and a constant minimum
   - The minimum is kept constant across images taken during the same imaging session and is determined as a mean minimum based on the intensities at control NMJs.
4. Particle analysis
   - Using ImageJ's 'Analyze Particles. . .' command
   - Determines number and average size of segmented puncta

## Electron microscopy

Third instar larvae were pinned down onto Sylgard-coated plates using 0.1 mm minutien pins (Fine Science Tools) and dissected in ice-cold 0.1M cacodylate buffer using techniques similar to those described above for immunostaining. Larval fillets were fixed with a solution of 2.5% paraformaldehyde, 5.0% glutaraldehyde, 0.06% picric acid in 0.1M cacodylate buffer, overnight in 4°C. Fillets were rinsed in 0.1M ice-cold cacodylate buffer, pH 7.4, then unpinned and trimmed for post-fix and embedding. Samples were post-fixed with 2% osmium tetroxide in 0.1M cacodylate buffer for 2 hr on ice, then rinsed with deionized water, then stained in 1% aqueous uranyl acetate for 2 hr. Fillets were dehydrated in graded alcohols and propylene oxide. Then, the samples were incubated in TAAB 812 resin (Canemco-Marivac, Quebec, Canada). Blocks were kept at 60°C for 48 hr to complete the polymerization process. Both semi and ultra-thin sections were prepared with Diatome Histo and Diatome Ultra 45° diamond knives respectively on Leica UC7 ultramicrotome. Sections were mounted on single slot grids with 2mmx1mm oval holes. Sections were imaged using AMT 2k CCD camera mounted on a Tecnai G$^2$ Spirit BioTWIN Transmission Electron Microscope (FEI Company) at 6800x magnification. Boutons were located based on morphological characteristics, in particular the presence of active zones / T-bars.

From the cross-sectional images of boutons, SSR widths were measured in ImageJ (NIH) as follows: (1) An ROI was drawn around the perimeter of the bouton, then he center of mass (i.e., center of bouton) was determined. (2) eight lines, 45 degrees apart were drawn emanating from the bouton's center. (3) On each of these lines, the length of the segment between the edge of the SSR and the edge of the bouton was measured, then averaged. All quantifications are shown in nm, with the mean and standard errors. Statistical significance was determined using a two-tailed Student's t test.

## Biochemical analysis

### Sample preparation

Third instar larvae were dissected at room temperature in PBS; gut and fat body were removed while the CNS was kept intact except for in *Figure 4C*, in which the CNS was isolated. Carcasses, now mostly body wall muscle (±CNS) in composition, were homogenized in 1X Laemmli buffer using a motorized Kontes pestle (Fischer Scientific). Samples were boiled for 5–10 min and stored in −20°C until ready to use. Number of larvae were kept consistent across a single experiment.

### Western blotting

SDS-PAGE and Western blotting were performed using standard protocols. A single % gel, usually at 12–15%, was used. All washes were done with a 0.05% PBS-Tween solution. For blocking and antibody incubation, 5% nonfat milk in a 0.05% PBS-Tween solution was used. Primary antibodies (below) were incubated overnight in 4°C; secondary antibodies (below) were incubated for 1 hr at room temperature. HRP-conjugated secondary antibodies were used for chemiluminescence, and

ECL reaction was performed using SuperSignal West Dura Extended Duration Substrate (Pierce Biotechnology, Thermo Scientific).

## Primary antibodies used in this study

| Antigen and clone ID | Species | Concentration | Reference / Source |
|---|---|---|---|
| Filamin (C terminus) (43-D) | Rabbit | 1:5000 | (*Li et al., 1999*) |
| Filamin (N terminus) | Rat | 1:2000 | (*Külshammer and Uhlirova, 2013*) / M. Uhlirova Lab |
| GluRIIA, concentrate | Mouse | 1:1000 | DSHB |
| HA (3 F10) | Rat | 1:1000 | Roche / RRID:AB_2314622 |
| Tubulin (DM1A) | Mouse | 1:10,000 | Sigma-Aldrich / RRID:AB_477583 |

## HRP-conjugated secondary antibodies used in this study

| Antigen | Concentration | Source |
|---|---|---|
| Mouse IgG light chain | 1:5000 | EMD Millipore / RRID:AB_805324 |
| Rabbit IgG (heavy + light chains) | 1:10,000 | Jakson Immunoresearch / RRID: AB_2307391 |
| Rat IgG (heavy + light chains) | 1:10,000 | Jakson Immunoresearch / RRID: AB_2340639 |

## Electrophysiology

Larvae were grown and collected as described above. third-instar larvae were dissected on Sylgard-coated plates in ice-cold Ca-free HL3 solution (*Stewart et al., 1994*), with the segmental nerves severed just before recording. Recordings were performed in HL3 solution with 1 mM $Ca^{+2}$, from muscles 6/7 in abdominal segments A2 and A3 using 10–20 mΩ sharp glass electrodes filled with 3M KCl. Miniature events were recorded at least 1 min after obtaining a stable membrane potential, in a 90 s period. For recording evoked potentials, severed segmental nerves innervating the respective muscles were stimulated using a suction electrode filled with the bath solution, via 0.2 Hz stimuli delivered through Clampex and the A365 stimulus isolator (WPI, Sarasota, FL). Only recordings with resting Vm < −55 mV were included for analysis. Data were collected via Axopatch 200B amplifier (Axon Instruments/Molecular Devices, Sunnyvale, CA) and Digidata (Molecular Devices), and processed using pClamp8 (Molecular Devices). mEJP and EJP parameters were detected under consistent settings using MiniAnalysis (Synaptosoft, Decatur, GA).

## In situ proximity ligation assay (PLA)

HEK293T (ATCC) cells, whose morphology was used for identification and are routinely used in the lab, were grown in DMEM (Dulbecco's Modified Eagle's Medium) and supplemented with L-glutamine, penicillin/streptomycin (ThermoFisher Scientific), and 10% fetal bovine serum (Atlanta Biologicals, Flowery Branch, GA), and, one day before transfection, plated on glass coverslips in a 24-well plate at a density of 0.15 million cells/mL. Constructs were transfected using standard calcium-phosphate protocol. Transgenes were under the control of a UAS promoter and co-transfected with a CMV-Gal4 construct. Cells were fixed with 4% paraformaldehyde in 1X PBS approximately 40 hr post transfection. PLA was performed using Duolink/PLA reagents (Sigma-Aldrich), according to the manufacturer's protocol. The blocking solution used was 5% normal goat serum in 0.3% Triton-X solution in 1X PBS. The following primary antibodies were used: mouse anti-V5 (Life Technologies (RRID:AB_2556564); at 1:500); rabbit anti-GFP (Life Technologies (RRID:AB_221569); at 1:1000). Cells were imaged using the Zeiss LSM700 confocal microscope with a 63x oil objective and images were acquired using the Zen software; GFP+ cells were counted manually and scored as PLA+ if they had more than three puncta of PLA reaction product.

## Constructs used for PLA

| Label | Description / reference |
|---|---|
| UAS-GFP-FLN90 | FLN90 isoform with an N terminus EGFP tag; see above for cloning details |
| UAS-V5-dPak | *Duan et al., 2012* (from E. Chen Lab; Johns Hopkins) |

## Number of animals used in quantified NMJ data

| Figure | Condition/genotype | # of NMJs (also indicated in each figure) | # of animals |
|---|---|---|---|
| 1F | RalGDP control | 14 | 4 |
| | RalGDP + filamin RNAi | 14 | 4 |
| | RalGTP control | 13 | 4 |
| | RalGTP + filamin RNAi | 12 | 4 |
| | HA-Ral control background | 24 | 6 |
| | HA-Ral mutant background | 25 | 7 |
| | Sec5 control | 17 | 11 |
| | Sec5 + filamin RNAi | 23 | 14 |
| Supp 1D | Control | 27 | 7 |
| | Filamin RNAi | 27 | 7 |
| 2B | Control | 54 | 16 |
| | filamin RNAi | 56 | 17 |
| | Control | 25 | 7 |
| | filamin mutant | 29 | 8 |
| 2C | Control | 45 | multiple |
| | filamin mutant | 53 | multiple |
| 2F | Control | 19 | 6 |
| | filamin RNAi | 23 | 6 |
| | filamin RNAi + RalGTP | 14 | 5 |
| 2H | Control | 16 | 7 |
| | filamin RNAi | 14 | 8 |
| | Control | 21 | 8 |
| | filamin mutant | 17 | 7 |
| Supp 2C | Control | 15 | 4 |
| | Ral RNAi | 12 | 4 |
| | Control | 6 | 2 |
| | Ral mutant | 9 | 3 |

| 3C | Synd - Control | 34 | 15 |
|---|---|---|---|
| | Synd - CherQ1415sd | 15 | 8 |
| | Synd - Cher1 | 36 | 11 |
| | Ral - Control (m67) | 81 | 27 |
| | Ral - CherQ1415sd (m67) | 25 | 7 |
| | Ral - Cher1 (m67) | 53 | 19 |
| | Ral - Control (m4) | 43 | 17 |
| | Ral - CherQ1415sd (m4) | 12 | 5 |
| | Ral - Cher1 (m4) | 39 | 14 |
| 3G | Control | 22 | 10 |
| | CherQ1415sd | 19 | 9 |
| | FLN90 rescue | 22 | 8 |
| 6B | Intensity - Control | 67 | 28 |
| | Intensity - Cher1 | 26 | 10 |
| | Intensity - CherQ1415sd | 45 | 18 |
| | Control - Intensity, puncta, size | 52 | 21 |
| | RNAi - Intensity, puncta, size | 62 | 28 |
| 6D | Intensity - Control | 14 | 7 |
| | Intensity - CherQ1415sd | 13 | 7 |
| | Control - Intensity, puncta, size | 13 | 10 |
| | RNAi - Intensity, puncta, size | 19 | 14 |
| 6F | Control - Intensity, puncta, size | 29 | 8 |
| | RNAi - Intensity, puncta, size | 22 | 7 |
| 6G | Control - Intensity, puncta, size | 27 | 7 |
| | RNAi - Intensity, puncta, size | 27 | 7 |
| Supp 6B | Control | 20 | 7 |
| | Ral null | 26 | 7 |
| 7B | mini freq and amplitude - control | 13 | 9 |
| | mini freq and amplitude - RNAi | 22 | 16 |
| 7D | EPSP amplitude and area- control | 7 | 5 |
| | EPSP amplitude and area - RNAi | 12 | 11 |
| 8B | Control | 44 | 17 |
| | dPak null | 22 | 8 |
| | No kinase domain | 12 | 4 |
| | Kinase dead | 13 | 7 |
| 8D | Control | 6 | 3 |
| | Ral RNAi | 6 | 3 |
| | Control | 11 | 3 |
| | Ral null | 6 | 2 |
| 8F | Control | 28 | 8 |
| | dPix null | 30 | 8 |
| 9A | Control | 15 | 4 |
| | RNAi | 11 | 3 |
| | Control | 12 | 4 |
| | CherQ1415sd | 10 | 4 |
| 9D | Control | 16 | 6 |
| | dPak RNAi | 15 | 7 |

| 9E | Control | 16 | 6 |
|----|---------|----|----|
|    | RNAi    | 20 | 7 |

## Acknowledgements

We thank members of the Schwarz laboratory in addition to Jonathan Cohen, Josh Kaplan, and Davie van Vactor for helpful discussion; Elizabeth Chen, Lynn Cooley, Aaron DiAntonio, Nicholas Harden, Thomas Hays, Matthew Pecot, Mani Ramaswami, Mirka Uhlirova, the Developmental Studies Hybridoma Bank, the *Drosophila* Genomics Resource Center, and the Bloomington *Drosophila* Stock Center for constructs, antibodies, and fly stocks; Guoli Zhao for technical assistance; Talya Kramer for assistance with fly husbandry; Grzegorz Gorski for assistance with sample preparation for electron microscopy; and the HMS Conventional Electron Microscopy Facility. This work was supported by National Institutes of Health R01 NS041062 to TLS and the IDDRC Cellular Imaging, Histology, and Molecular Genetics Cores (NIH P30 HD018655).

## Additional information

### Funding

| Funder | Grant reference number | Author |
|--------|------------------------|--------|
| National Institutes of Health | R01 NS041062 | Thomas L Schwarz |
| National Institutes of Health | NIH P30 HD018655 | Thomas L Schwarz |

The funders had no role in study design, data collection and interpretation, or the decision to submit the work for publication.

### Author contributions

GYL, Conception and design, Acquisition of data, Analysis and interpretation of data, Drafting or revising the article, Contributed unpublished essential data or reagents; TLS, Conception and design, Analysis and interpretation of data, Drafting or revising the article, Contributed unpublished essential data or reagents

### Author ORCIDs

Thomas L Schwarz, http://orcid.org/0000-0001-7532-0250

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
