## [Decision Letter]

Thank you for submitting your article "*Filamin*, an organizer of the *Drosophila* NMJ, determines glutamate receptor composition and postsynaptic membrane growth" for consideration by *eLife*. Your article has been reviewed by three peer reviewers, one of whom is a member of our Board of Reviewing Editors, and the evaluation has been overseen by K VijayRaghavan as the Senior Editor. The reviewers have opted to remain anonymous.

The reviewers have discussed the reviews with one another and the Reviewing Editor has drafted this decision to help you prepare a revised submission.

Summary:

Lee and Schwarz present a study of molecular interactions governing maturation of the *Drosophila* larval neuro-muscular junction (NMJ), a well-established synaptic model. The Schwarz group had previously shown that synaptic localization of the small GTPase Ral (and subsequent recruitment of the exocyst complex) is critical for proper formation of the subsynaptic reticulum, a major postsynaptic structure. In the current study they demonstrate that proper Ral localization at the NMJ synapse depends on a pathway involving the scaffold/actin-binding protein Filamin-A/Cheerio (FLN-A) and the kinase dPak and associated elements, with FLN-A acting as the most upstream element. Remarkably, the FLN-A protein involved is a short isoform (FLN90) lacking the actin-binding domain. The FLN-dPak pathway is also shown to (separately) influence the composition of postsynaptic glutamatergic receptors, by ensuring specific recruitment of type A glutamate receptors. This study identifies FLN90 as a "strategic" mediator of key morphological and functional aspects of the postsynaptic portion of the NMJ.

Essential revisions:

All three reviewers thought that the paper addressed important subject matter and found the experimental work to be of high quality. Reservations were raised, however, regarding how well the molecular model at the heart of the study (Figure 9) is substantiated by the data, and regarding the degree of mechanistic insight obtained. Additional supporting data and text revisions along three major lines of investigation are therefore deemed necessary and should be pursued:

1) The entire analysis of post-synaptic phenotypes and localization patterns appears to have been performed on third instar larvae, an advanced stage of larval NMJ morphogenesis. Given the profound effects of loss of filamin function, specifically on the SSR, it would be expected that practically any SSR-associated protein will be delocalized under these circumstances, making it difficult to distinguish between filamin-dependent recruitment and secondary localization effects. The authors should therefore begin to evaluate the localization of dPak and Ral (in wildtype and filamin loss-of-function backgrounds) from the earliest stages of SSR formation, in order to better establish the recruiting role of Filamin (see for example the analysis of Dlg function during SSR development (Guan et al. Current Biology (1996) 6:695-706)).

2) The proposed role of Filamin as a physical scaffold for key elements mediating NMJ morphogenesis rests primarily on mutant analysis and genetic interactions. Have the authors made any efforts to look for complexes between Filamin and Ral/dPak by co-IP, proximity ligation assay, etc.? Data of this type could provide important insight regarding the mechanistic basis of Filamin function.

3) The data presented suggests that the short Filamin isoform FLN90 is the relevant form for NMJ functions. This has important mechanistic significance, as FLN90 lacks the conserved actin-binding domain. While considerable evidence is presented in support of this assertion, it is still circumstantial, and the authors should carry out a definitive rescue experiment- namely use the GAL4-UAS system to show that (postsynaptic) FLN90 rescues the filamin null phenotype.

---

## [Author Response]

**[…]**

*Essential revisions:*

*All three reviewers thought that the paper addressed important subject matter and found the experimental work to be of high quality. Reservations were raised, however, regarding how well the molecular model at the heart of the study (Figure 9) is substantiated by the data, and regarding the degree of mechanistic insight obtained. Additional supporting data and text revisions along three major lines of investigation are therefore deemed necessary and should be pursued:*

*1) The entire analysis of post-synaptic phenotypes and localization patterns appears to have been performed on third instar larvae, an advanced stage of larval NMJ morphogenesis. Given the profound effects of loss of filamin function, specifically on the SSR, it would be expected that practically any SSR-associated protein will be delocalized under these circumstances, making it difficult to distinguish between filamin-dependent recruitment and secondary localization effects. The authors should therefore begin to evaluate the localization of dPak and Ral (in wildtype and filamin loss-of-function backgrounds) from the earliest stages of SSR formation, in order to better establish the recruiting role of Filamin (see for example the analysis of Dlg function during SSR development (Guan et al. Current Biology (1996) 6:695-706)).*

We thank the reviewers for pointing out an important aspect of interpreting our data. In addition to clarifying what had already been presented in the earlier version regarding this issue, we now present additional pieces of data that further clarify the relationship between protein localization and SSR formation. We now cite our earlier paper in which we had shown that subsynaptic Ral localization did not require the formation of the SSR (Teodoro et al., 2013) and also offer new evidence that this is true for dPak as well (this manuscript, now Figure 8). We also now show that this is the case for FLN90, by demonstrating its subsynaptic localization in a ral-/- mutant background that lacks SSR (Figure 5). Together these experiments demonstrate that Filamin, dPak, and Ral are required for SSR formation but do not require the SSR to form for their own localization. dPak and Ral localization are directly dependent on filamin and their mislocalization in filamin mutants are not an indirect effect of SSR loss. Furthermore, the synaptic presence of FLN90 in ral-/- background makes the case for FLN90 being present prior to and driving SSR growth.

*2) The proposed role of Filamin as a physical scaffold for key elements mediating NMJ morphogenesis rests primarily on mutant analysis and genetic interactions. Have the authors made any efforts to look for complexes between Filamin and Ral/dPak by co-IP, proximity ligation assay, etc.? Data of this type could provide important insight regarding the mechanistic basis of Filamin function.*

Yes, we have indeed taken the biochemical approach to supplement the genetic arguments. Attempts to co-IP from larvae or after co-expression in cultured cells were frustrated by technical difficulties including a non-specific affinity of filamin for beads. We very much thank the reviewers, however, for suggesting proximity ligation assay (PLA) as an alternative. We have taken that suggestion and carried out a proximity ligation assay that corroborated the data from mammalian studies by indicating that FLN90 and dPak are found in complex (new Figure 9). We explain in the text that we chose to do this assay by co-expression in HEK cells rather than in situ at larval NMJs because PLA tests for proximity within 40nm. Because the folds of the SSR and the postsynaptic density are so dense, it seemed likely we could get a positive signal from PLA even if there was no direct interaction of the two proteins. The positive result from HEK cell expression therefore (and negative result with an appropriate control) are therefore a more robust indication of a genuine interaction and we are indebted to the reviewers for this useful strategy for augmenting the manuscript.

*3) The data presented suggests that the short Filamin isoform FLN90 is the relevant form for NMJ functions. This has important mechanistic significance, as FLN90 lacks the conserved actin-binding domain. While considerable evidence is presented in support of this assertion, it is still circumstantial, and the authors should carry out a definitive rescue experiment- namely use the GAL4-UAS system to show that (postsynaptic) FLN90 rescues the filamin null phenotype.*

We have added the requested rescue experiment (new Figure 3) which shows that muscle-specific expression of FLN90 in a null background is sufficient, as indicated by the previous allelic analysis and immunocytochemistry of the two isoforms.